# The solution structure of Dead End bound to AU-rich RNA reveals an unusual mode of tandem RRM-RNA recognition required for mRNA regulation

Malgorzata M. Duszczyk[1] ✉, Harry Wischnewski[2], Tamara Kazeeva[1], Rajika Arora [2], Fionna E. Loughlin [1,5], Christine von Schroetter[1,3], Ugo Pradère[4], Jonathan Hall [4], Constance Ciaudo [2] ✉ & Frédéric H.-T. Allain [1,3] ✉

Dead End (DND1) is an RNA-binding protein essential for germline development through its role in post-transcriptional gene regulation. The molecular mechanisms behind selection and regulation of its targets are unknown. Here, we present the solution structure of DND1's tandem RNA Recognition Motifs (RRMs) bound to AU-rich RNA. The structure reveals how an NYAYUNN element is specifically recognized, reconciling seemingly contradictory sequence motifs discovered in recent genome-wide studies. RRM1 acts as a main binding platform, including atypical extensions to the canonical RRM fold. RRM2 acts cooperatively with RRM1, capping the RNA using an unusual binding pocket, leading to an unusual mode of tandem RRM-RNA recognition. We show that the consensus motif is sufficient to mediate upregulation of a reporter gene in human cells and that this process depends not only on RNA binding by the RRMs, but also on DND1's double-stranded RNA binding domain (dsRBD), which is dispensable for binding of a subset of targets in cellulo. Our results point to a model where DND1 target selection is mediated by a non-canonical mode of AU-rich RNA recognition by the tandem RRMs and a role for the dsRBD in the recruitment of effector complexes responsible for target regulation.

Post-transcriptional gene regulation (PTGR) is orchestrated by an interplay between mRNA sequence and structure, and their dynamic interactions with RNA-binding proteins (RBPs). RBPs instantly cover mRNA transcripts as they are transcribed and are essential for all aspects of RNA metabolism like maturation, transport, cellular localization, and turnover. Differential gene expression patterns depend on tissue-specific RBP levels and their combined interactions with the transcriptome. Misregulation of this process, due to mutations in the

RBPs or RNA to which they bind, is at the origin of a plethora of genetic diseases[1]. Understanding how RBPs specifically recognize their mRNA targets and how this is connected to their downstream fate is therefore crucial to understand the complex PTGR networks involved in health and disease.

The germline is composed of highly specialized cells which must preserve a totipotent genome through generations. It succeeds in this through a highly specialized RNA metabolism regulating a complex

[1]Institute of Molecular Biology and Biophysics, ETH Zürich, 8093 Zürich, Switzerland. [2]Institute of Molecular Health Sciences, ETH Zürich, 8093 Zürich, Switzerland. [3]Institute of Biochemistry, Department of Biology, ETH Zürich, 8093 Zürich, Switzerland. [4]Institute of Pharmaceutical Sciences, Department of Chemistry and Applied Biosciences, ETH Zürich, 8093 Zürich, Switzerland. [5]Present address: Monash Biomedicine Discovery Institute, Department of Biochemistry and Molecular Biology, Monash University, Clayton 3800 VIC, Australia. ✉e-mail: m.m.duszczyk@gmail.com; cciaudo@ethz.ch; allain@bc.biol.ethz.ch

transcriptome[2,3]. Dead End (DND1) is a germline-specific RBP. Conserved in vertebrates, it is essential for the development and migration of primordial germ cells (PGCs), pluripotent germline precursors, to the future reproductive organs. These processes occur early in embryogenesis by blocking the expression of somatic genes, controlled by extensive post-transcriptional regulation[3,4]. DND1 deletion causes loss of PGCs by reactivation of somatic gene expression patterns in zebrafish[5,6]. In mice, truncations of DND1 (the so-called "Ter-mutation") lead to male sterility and the formation of testicular teratomas[7,8].

DND1 performs a dual role in regulating gene expression: stabilizing some mRNA transcripts while destabilizing others. First, a direct role for DND1 in promoting RNA stability by inhibiting the action of miRNAs was shown for specific candidate mRNAs (p27/CDKN1B and LATS2 in a human tumor cell line and Nanos/TDRD7 in zebrafish embryos)[9]. DND1 binding to conserved U-rich regions close to miRNA seed sequences in the 3′ untranslated regions (3′UTRs) of these targets potentially blocked their accessibility to the miRNA-induced silencing complex (miRISC) which rescued translation. Second, on a transcriptome-wide level, DND1 was shown to have an opposite effect and a wider role in germ cell PTGR by destabilizing a set of transcripts that must be cleared from developing germ cells to ensure survival, through non-miRISC mediated recruitment of the CCR4-NOT deadenylase complex[10,11].

Transcriptome sequencing of the Ter mouse line, possessing a truncation in the DND1 gene, prior to the formation of teratomas, showed two groups of DND1 targets either up- or downregulated compared to wild-type, involved in pluripotency and in differentiation, respectively, in a sequential fashion, suggesting a dual and biphasic role for DND1. No enrichment for RNAs harboring a signature for miRNA regulation was found in the wild-type versus Ter cell line, although previously described DND1 targets were re-identified[7].

Photoactivatable-ribonucleoside-enhanced cross-linking and immunoprecipitation (PAR-CLIP) assays revealed that targets cross-linked to DND1 are enriched in a UUU/UUA triplet and are associated with apoptosis, inflammation, and signaling pathways[11]. An additional digestion optimized-RIP-seq (DO-RIP-seq) approach described [A/G/U]AU[C/G/U]A[A/U] as RNA sequence motif enriched in DND1 targets[7]. Overall, these functional and genome-wide motif identification studies using high-throughput methods are partly contradictory. Thus, the molecular mechanisms of DND1 function and details on how DND1 achieves target specificity remain elusive.

Most RNA-binding proteins exert their specific functions by combining several copies of RNA-binding domains to increase specificity and affinity to their targets[12,13]. DND1 has a unique domain structure, carrying two RNA recognition motifs (RRMs) connected by a remarkably short, four-residue interdomain linker preceding a double-stranded RNA-binding domain (dsRBD) (Fig. 1A). This combination of domains is rare among RBPs and is only shared by DND1 and two other members of the hnRNPR-like subfamily[14] (Supplementary Fig. 1A, C, E). RRMs are the most abundant RNA-binding domains in higher vertebrates, binding primarily single-stranded RNA[15,16]. Their conserved canonical fold is βαββαβ—a four-stranded anti-parallel β-sheet packed on top of two alpha-helices. Their canonical mode of RNA binding involves stacking of two consecutive RNA bases by exposed aromatic residues from the two conserved consensus sequences (RNP1 & RNP2) in the first and third β-strands (Supplementary Fig. 1B). The RRM uses several strategies to achieve specificity and affinity for its targets, namely by using extensions to the canonical fold with structural elements outside the β-sheet for RNA recognition and by employing several copies of the domain. While DND1's RRM1 appears to be a canonical RRM, RRM2 on the contrary does not have any aromatic residues in RNP1 and RNP2 (Fig. 1A and Supplementary Fig. 1B, C). Although several structures of tandem RRM-RNA complexes have been determined[17-22], the great majority of them contain combinations

of two canonical RRMs. It is therefore intriguing to understand if and how the tandem RRMs of DND1 cooperate to specifically recognize their RNA targets, and if the dsRBD further influences RNA binding.

To address the question of how DND1 recognizes and represses its cellular targets at the molecular level, we first set out to understand the contribution of the three RNA-binding domains of DND1 to target recognition. This demonstrated a crucial role for the RRMs. We then determined the solution structure of the DND1 tandem RRMs in complex with an AU-rich RNA. Our structure reveals an unusual mode of cooperative binding by tandem RRMs and explains the degeneracies in the motifs enriched in DND1 targets in recent genome-wide studies.

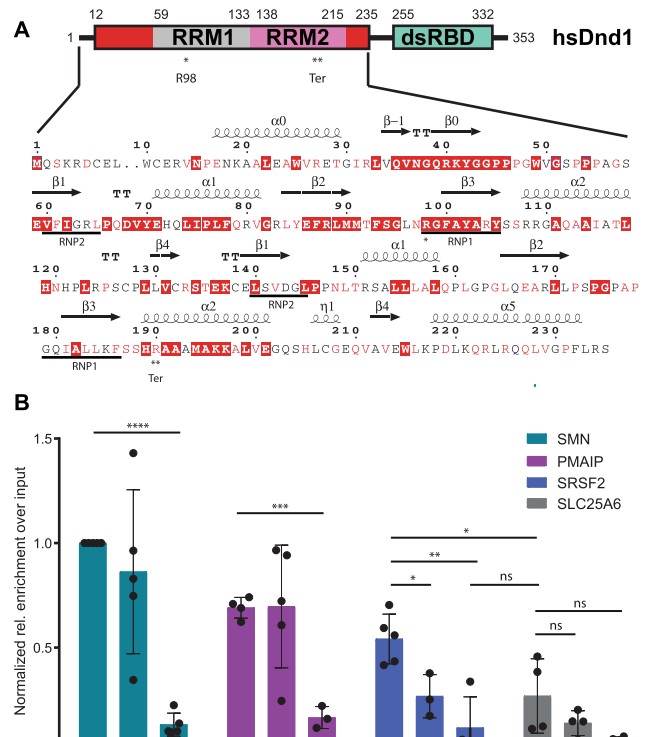

**Fig. 1 | DND1 binds RNA targets mainly through its RRMs. A** Domain structure of DND1 and sequence of the N-terminal part of the human protein (Uniprot Q8IYX4 [https://www.uniprot.org/uniprotkb/Q8IYX4/entry]) ending at the C-terminus of the tandem RRM construct used for structure determination in this work (12–235). RRM1 in gray, RRM2 in pink, N- and C-terminal conserved extensions in red, dsRBD in green. The dsRBD-truncation mutant 1-235 used in our RIP assay ends after the extension of RRM2. Red coloring in the sequence indicates high conservation as described in Supplementary Fig. 1C. Secondary structure elements as found in our structure are indicated above the sequence. The RRM-canonical RNA-binding RNP sequences are underlined below. R98 in RNP1 that was mutated for the RIP assay is indicated with one asterisk. The Ter truncation at R190 is indicated with a double asterisk and "Ter". See also Supplementary Fig. 1A, C, **B** RNA Immunoprecipitation from HEK293T cells transiently expressing FLAG-tagged DND1 or its mutants followed by qPCR using primers for published DND1 targets and negative control (Supplementary Table 1). Data from five independent experiments is presented as relative enrichment over the input ($2^{-\Delta Ct}$), normalized to the enrichment of the SMN targets pulled down by Dnd1 WT. ΔCt is an average of (Ct [RIP] − (Ct [Input]) of technical triplicates with SD < 0.3. If SD (technical triplicate) was > 0.3 the data point was omitted. Only data with at least $N = 3$ is presented. The results are represented as means and SD. $P$ values from two-tailed Welch's $t$-test: *$P < 0.05$ (exact values 0.0191 for SRSF2 1-235 vs WT; 0.0460 for SRSF2 WT vs SLC25A6 WT); **$P < 0.01$ (exact value 0.0038 for SRSF2 R98A vs WT); ***$P < 0.001$ (exact value 0.0001 for PMAIP R98A vs WT); ****$P < 0.0001$. DND1 and mutants are well expressed in HEK293T cell culture (Supplementary Fig. 2). Source data are provided as a Source Data file.

Using luciferase-based assays, we validated not only the role of the tandem RRM-AU-rich RNA interaction, but in addition also could pinpoint an essential role for the dsRBD, in gene expression regulation by DND1. Finally, immunoprecipitation followed by liquid chromatography-tandem mass spectrometry analysis of the interactome of DND1 targets in HEK293T reveals enrichment for proteins involved in the stabilization of mRNAs like ELAVL1 and proteins localized in RNP granules. These results provide the first mechanistic and structural insights into the molecular mechanisms by which DND1 regulates a subset of mRNAs, thereby stabilizing the fate of germ cells.

## Results

### DND1 binds CLIP/RIP targets in cellulo mainly through its RRM1

There is some ambiguity in the published RNA motifs targeted by DND1. It was first reported on the basis of data from reporter assays, that DND1 binds to U-rich regions of ~12 nucleotides in length (UUUUUCCUUAUUU and UUUUUACCUUUU) in the vicinity of miRNA seed sequences in the 3'UTR of the CDKN1B/p27 tumor suppressor mRNA[9]. Later, genome-wide PAR-CLIP studies defined a much shorter UUU/UUA triplet motif[11] and very recently a Digestion Optimized-RIP approach revealed [A/G/U]AU[C/G/U]A[A/U] (DAUBAW) as motif enriched in DND1 targets[7]. A single RRM usually binds 4–6 nucleotides[15]. To understand how the two RRMs of DND1 cooperate with the dsRBD to engage an RNA target, we set out to define the contributions of each domain to RNA binding. We first validated published DND1 targets using RNA immunoprecipitation (RIP) from HEK293T cells transiently expressing either FLAG-tagged DND1 wild-type (WT) or mutants thereof. Mutant 1-235 lacks the dsRBD but includes the extended RRM2, making it longer than the Ter-mutation truncation[8], which is located in the middle of RRM2 α-helix 2 (Fig. 1A); R98A contains a mutation of the conserved RNP1 sequence in RRM1 (Fig. 1A and Supplementary Fig. 1B). The RIP was followed by quantitative PCR using primers for two DND1 targets revealed by PAR-CLIP and DO-RIP (Fig. 1B). Phorbol-12-Myristate-13-Acetate-Induced Protein 1 (PMAIP1), a cell cycle regulator promoting apoptosis, is the target with the highest number of normalized 3'UTR read counts in the PAR-CLIP dataset[11]. It contains 15 UAUU motifs in its 3'UTR and was previously identified by DO-RIPs[7]. Serine/arginine-rich splicing factor 2 (SRSF2) is a member of the SR protein family that is involved in both constitutive and alternative mRNA splicing. It had the third highest number of normalized 3'UTR read counts in the PAR-CLIP dataset[11] and is expressed at the same level as PMAIP. It contains 15 UAUU motifs in its 3'UTR and was previously identified by DO-RIPs[7]. Survival of Motor Neuron (SMN) protein is a small nuclear ribonucleoprotein (snRNP) assembly factor. Its pre-mRNAs SMN1 and SMN2 are expressed at an order of magnitude lower level than PMAIP1 in HEK293, contain several UAUU motifs in the 3'UTR, depending on the transcript variant and are enriched in the DO-RIP dataset[7]. They had no reads in the PAR-CLIP dataset[11]. As a negative control, we used solute carrier family 25 member 6 (SLC25A6), the mRNA of which is expressed at similar levels as PMAIP1 in HEK293T cells[11] and had no reads in either DO-RIP[7] or PAR-CLIP[11] datasets, even though it contains one UAUU motif in its 3'UTR. Interestingly, in five independent RNA-IP followed by RT-qPCR experiments, the SMN1/2 mRNAs showed the highest enrichment over the input when pulled down by WT DND1, even though the expression in HEK293 is an order of magnitude lower than PMAIP and SRSF2 and the SMN targets are missing in the PAR-CLIP dataset[11]. All experiments were normalized to the enrichment of SMN by DND1 WT pulldown (Fig. 1B). While the SMN and PMAIP targets show no significant difference in enrichment over the input between the WT DND1 and the 1-235 dsRBD-truncation mutant, for SRSF2 the dsRBD does contribute to target binding (Fig. 1B). For all three targets, the full-length R98A mutant shows a highly significant reduction in enrichment over the input. The level of target enrichment by R98A pulldown is comparable to the negative control SLC25A6 pulled down by the WT

DND1. These data suggest that RRM1 is an essential element for all RNA target binding in cells, while the dsRBD appears to be dispensable for RNA binding of a subset of abundantly expressed targets, while it contributes to the binding of a third target. The role of RRM2 could not be tested easily, as its RNA interaction surface could not be predicted considering the noncanonical nature of this RRM. Therefore, we next investigated the contribution of the individual RRMs to RNA binding in vitro.

### DND1's tandem RRMs cooperatively bind AU-rich RNA with high affinity

To identify high-affinity RNA targets for the individual RRMs, we recombinantly expressed and purified the separate RRM domains, including their conserved N- and C-terminal extensions (Fig. 1A and Supplementary Fig. 1C). We then tested their binding to short oligonucleotides derived from the p27 target[9] using isothermal titration calorimetry (ITC) to monitor thermodynamic changes upon ligand binding (Fig. 2). In addition, we used NMR titrations− series of two-dimensional $^1H$-$^{15}N$ HSQC spectra of protein with increasing RNA concentrations−to observe perturbations of backbone amide proton chemical shifts caused by RNA binding (Fig. 2B and Supplementary Fig. 3). We found that RRM2 alone does not bind any of the oligonucleotides (Supplementary Fig. 3A) and that only an oligonucleotide bearing a central adenosine (UUAUUU) has sufficient affinity to RRM1 to yield significant chemical shift perturbations in an NMR titration (Supplementary Fig. 3B). The binding affinity of a slightly longer oligonucleotide CUUAUUUG to RRM1 is higher than 54 µM as measured by ITC (Fig. 2C). Surprisingly, the affinity for this sequence increased over 80-fold (0.68 µM $K_D$) when titrated with the tandem RRM domains showing the importance of RRM2 for RNA binding (Fig. 2A). Additional ITC binding experiments of RRM1 and RRM12 to AU-rich RNA are shown in Supplementary Fig. 3. NMR titration of CUUAUUUG to DND1−RRM12 confirmed the increased affinity with a change from fast to intermediate exchange on the chemical shift timescale, and the saturation of most residues at a 1:1 ratio (Fig. 2B). Large chemical shift changes throughout the $^1H$-$^{15}N$ HSQC spectrum indicated major structural rearrangements of the tandem RRMs upon RNA binding. Additional ITC and NMR titrations (Fig. 2D, E and Supplementary Fig. 3) showed that only the tandem RRMs, not the single domains, have affinity to U-rich oligonucleotides devoid of adenosines, or with an adenosine at the periphery such as UUUUUAC (>35 µM $K_D$) and UUUUUCC (>100 µM $K_D$). The in vitro binding measurements are summarized in Supplementary Table 2. Taken together, the results above indicated that the DND1 tandem RRMs cooperatively bind AU-rich RNA targets of ~7–8 nucleotides in length, with the highest affinity to oligonucleotides comprising an adenosine in a central position.

### The solution structure of DND1's tandem RRMs bound to CUUAUUUG RNA

To understand the mechanism of cooperative binding of the tandem RRMs of DND1, as well as their preference for AU-rich over U-rich RNAs, we solved the structure of the extended RRM12 in complex with the CUUAUUUG RNA target using NMR spectroscopy (Supplementary Table 3 and Fig. 3). The assignment procedure of protein and RNA and the structure calculation protocol is outlined in the Methods section. DND1−RRM12 showed poor solubility and, in the final stages of purification, was purified as a complex with the target RNA. The use of chemically-synthesized CUUAUUUG oligonucleotides, which were selectively labeled with $^{13}C$-ribose units, was key to resolving the spectral overlap of critical residues and aided the assignments of resonances and intermolecular NOEs (Supplementary Fig. 4A). Hence, we could calculate a precise structural ensemble of this 27.5 kDa protein−RNA complex using unambiguous intraprotein (4947), intra-RNA (150) and intermolecular (103, Supplementary Data 2) NOE-derived distance restraints. The isolated domains within this ensemble

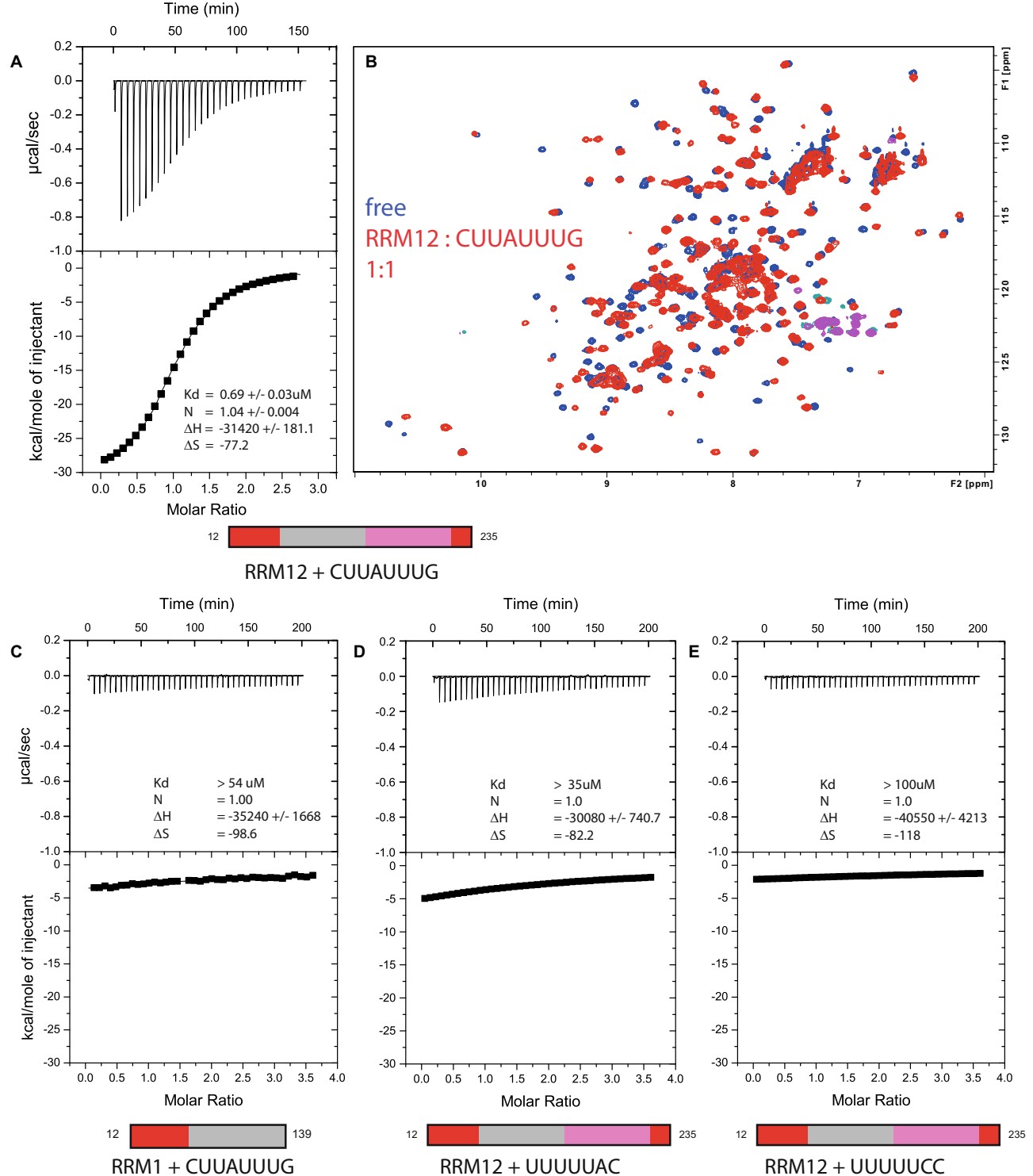

**Fig. 2 | Cooperative binding of DND1's tandem RRMs to AU-rich RNA. A** ITC measurement of DND1's tandem RRM domains titrated with CUUAUUUG RNA. $N = 1.04 +/- 0.004$; $K_D = 0.69 +/- 0.03$ uM; $\Delta H = -31.4 +/- 0.2$ kcal/mol; $-T\Delta S = 23.0$ kcal/mol. **B** overlay of two $^1H$-$^{15}N$ HSQC spectra of the free DND1 tandem RRMs (in blue) and a 1:1 complex with CUUAUUUG RNA (in red) demonstrates cooperative RNA binding. NMR conditions: Protein concentration 0.2 mM, 298 K, 750 MHz. NMR/ITC buffer: 100 mM Potassium Phosphate pH 6.6, 1 mM DTT. **C** ITC measurement of DND1's extended RRM1 titrated with CUUAUUUG ($N = 1$ (set); $K_D > 54$ µM; $\Delta H = -35 +/- 2$ kcal/mol; $-T\Delta S = 29.4$ kcal/mol). **D** ITC measurement of DND1's tandem RRMs titrated with UUUUUAC ($N = 1$ (set); $K_D > 35$ µM; $\Delta H = -30.1 +/- 0.7$ kcal/mol; $-T\Delta S = 24.5$ kcal/mol) and **E** DND1's tandem RRM domains titrated with UUUUUCC ($N = 1$ (set); $K_D > 100$ µM; $\Delta H = -41 +/- 4$ kcal/mol; $-T\Delta S = 35$ kcal/mol). See also Supplementary Fig. 3 and Supplementary Table 2.

of 20 conformers converge to a root mean square deviation (RMSD) of 0.95 Å (RRM1), 0.57 Å (RRM2) and the $U_3A_4U_5U_6U_7$ RNA nucleotides to an RMSD of 0.6 Å for all heavy atoms (Fig. 3A, B and Supplementary Table 3). The global complex was initially less well-defined due to a lack

of direct contacts between RRM1 and RRM2 and a limited number of restraints that could orient the domains through the RNA. We expected a well-defined orientation between RRM1 and RRM2 in the complex though, as $^{15}N$ T1 and T2 relaxation measurements indicated that the

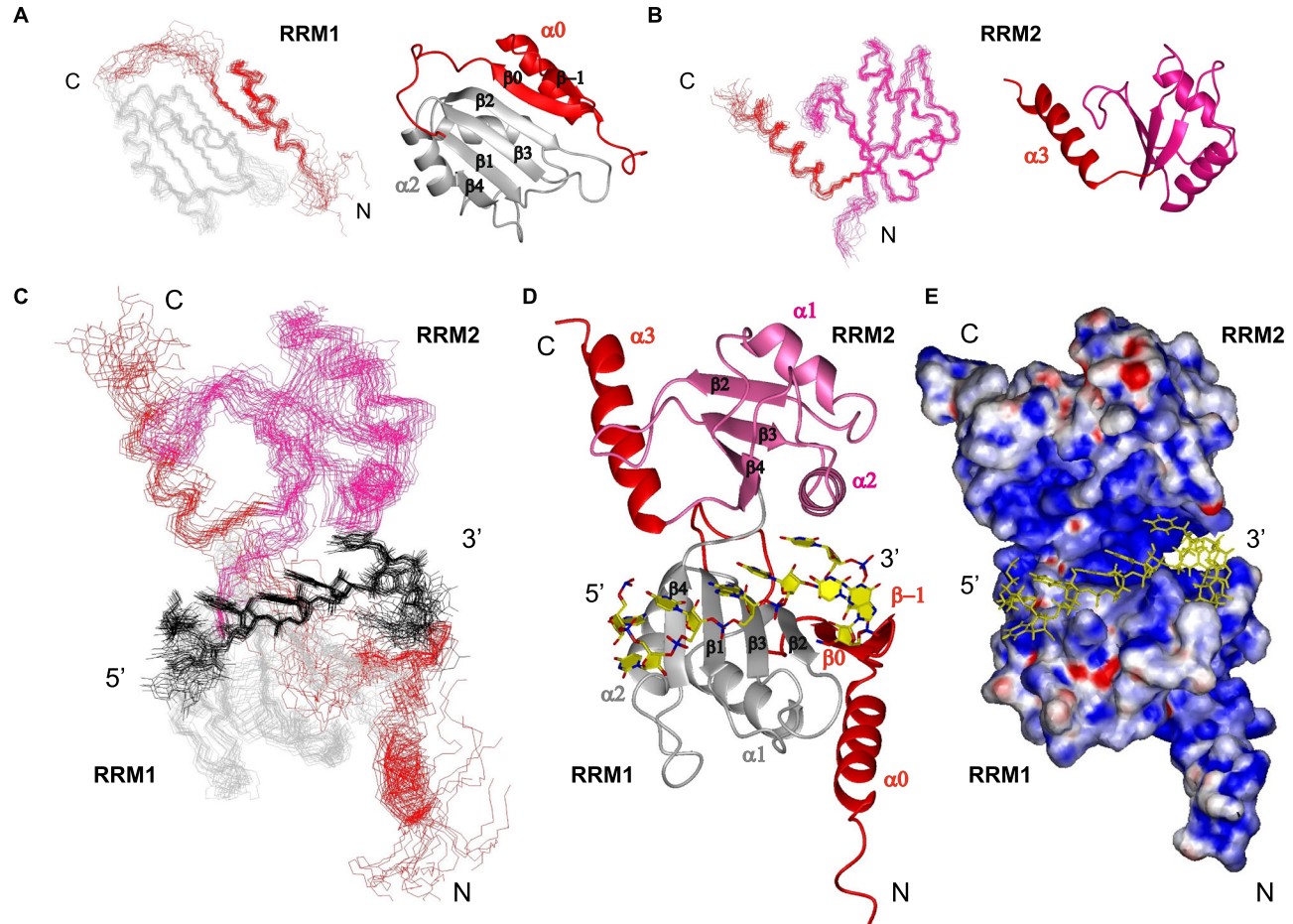

**Fig. 3 | Solution structures of DND1's tandem RRMs bound to CUUAUUUG RNA.** A–C Ensembles of 20 lowest energy conformers. RRM1 in gray, RRM2 in pink, conserved N- and C-terminal extensions in red. RNA in black. **A** Superimposed 20 lowest energy conformers and representative structures for RRM1 and **B** RRM2 within the tandem RRM-RNA complex. **C** Ensemble of 20 superimposed lowest energy conformers of the tandem RRM-RNA complex. **D** Representative structure, color coding protein as in **C**, RNA in yellow. **E** Electrostatic surface potential plot shows the RNA is buried in a positively charged cleft between the RRMs. See also Supplementary Fig. 4 and Supplementary Table 3.

protein–RNA complex behaves as a rigid globular protein of ~30 kDa in solution, as opposed to two domains tumbling independently (Supplementary Fig. 4B). We included 127 Residual Dipolar Couplings (RDCs) from amide bonds to increase precision and finally obtained a final ensemble of 20 conformers with an RMSD of 1.26 Å for all heavy atoms (Fig. 3C and Supplementary Table 3).

Our structure reveals a number of unusual structural features, including distinctive modes of RRM:RNA recognition. The fold of RRM1 is noncanonical with the conserved N-terminal extension folding into a β-hairpin (β−1,0) packed on top of an N-terminal α-helix (α0). This structural element is tightly packed against the side of the RRM1 to create an extended surface (Fig. 3A), in an identical fashion as the extension of RRM1 in the RNA-binding protein hnRNPQ/SYNCRIP[23]. This extended RRM (eRRM) fold is conserved in all members of the hnRNPR-like family of RNA-binding proteins (Supplementary Fig. 1A, D). RRM2 precedes the conserved C-terminal helical extension (α3) lacking contacts to the core canonical RRM2 fold (Fig. 3B), as confirmed by the relaxation data which showed this helix tumbling independently in the tandem RRM-RNA complex (Supplementary Fig. 4B). The RNA is bound in a canonical 5′ to 3′ fashion over β4 through to β2 using the RNP residues of RRM1, but is sandwiched between the two domains (Fig. 3D), burying the central UAUUU nucleotides in a positively charged channel (Fig. 3E). In addition to the primary RNA-binding surface on RRM1, the conserved N-terminal hairpin extension is used to extend RNA binding for a longer sequence compared to

canonical RRMs. Finally, and most surprisingly, this RNA binding is stabilized by RRM2 using an unusual binding pocket formed by RRM2 α-helix 2 and the edge of its β4, while the noncanonical β-sheet of RRM2, missing conserved aromatic RNP residues, is turned away from the RNA. Additional protein–RNA contacts are facilitated by the backbone and sidechains of the four-residue interdomain linker. This structure explains well the increase in RNA-binding affinity of the tandem RRMs compared to RRM1 in isolation.

**Structural details: specific readout by DND1's tandem RRMs**

DND1's tandem RRMs directly bind primarily the central 5 nucleotides (bold) of CU**UAUUU**G RNA (Fig. 4). The intermolecular contacts are represented in a schematic fashion in Fig. 4A. $U_3$ and $A_4$ are bound in a canonical fashion over the RRM1 β-sheet surface, their bases interacting with β4 and β1 respectively. The $U_3$ O2 is hydrogen-bonded to the C132 sidechain, and its ribose contacts L130. This contact is probably not fully specific, since a cytidine would also be accepted in this position keeping this H-bond intact. The $A_4$ base is stacked on F61 in RNP2 and its ribose interacts with F100 from RNP1 in a canonical manner (Fig. 4B). This is suggested by the unusual chemical shifts of the carbons and protons of the $A_4$ ribose (Supplementary Fig. 4A). The $A_4$ base is sequence-specifically recognized via hydrogen bonds to its Hoogsteen edge (N6 amino and N7) from sidechains and main chains of the highly conserved interdomain linker (R133, S134, and T135) (Fig. 4B, C). We observed

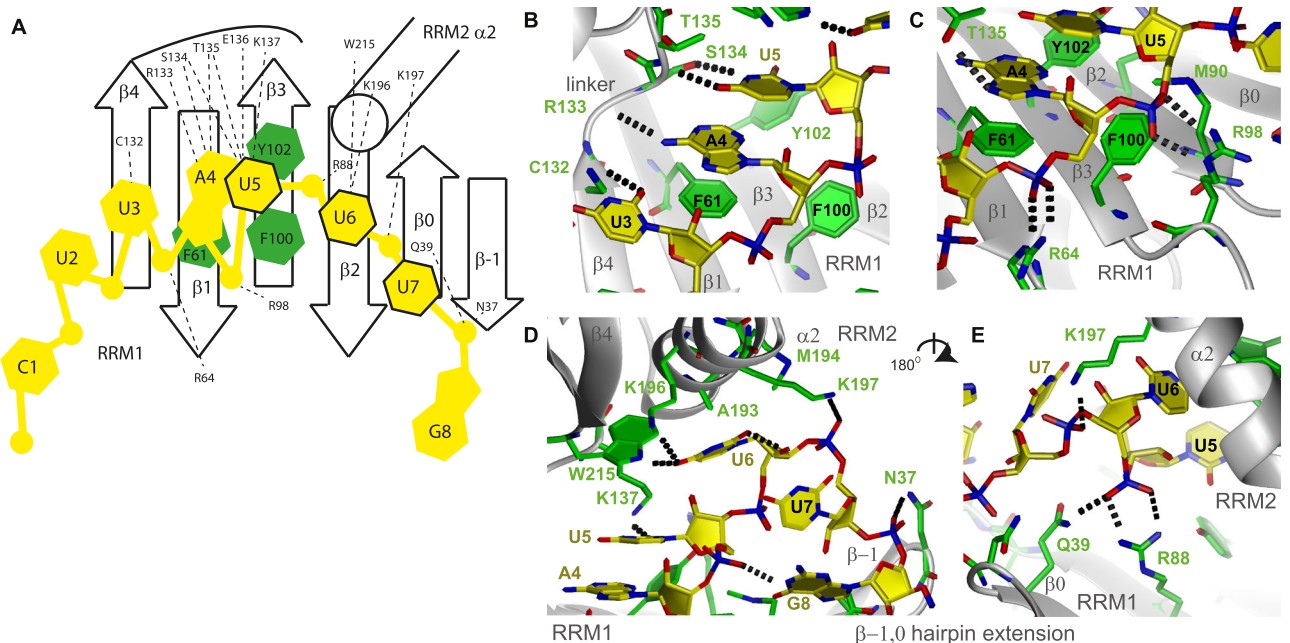

**Fig. 4 | Intermolecular contacts between the DND1 tandem RRMs and the CUUAUUUG RNA. A** Schematic view of protein–RNA interactions. **B** $U_3$, $A_4$, and $U_5$ base moieties. **C** $A_4$ and $U_5$ backbone. **D** $U_5$, $U_6$, and $U_7$ binding to the interdomain linker and the RRM2-binding pocket. $G_8$ binding to the eRRM1 β-hairpin extension.

**E** Cooperative binding by RRM1 and the $U_6$ and $U_7$ phosphate backbone, seen from the back of the molecule. Protein sidechains in green, RNA in yellow. Hydrogen bonds in dots. See also Supplementary Fig. 5.

variability in the H-bonding partners of $A_4$ in the structural ensemble, which reflects the exchange-broadening observed for the backbone resonances of these linker residues.

From $U_5$ onwards, the protein–RNA interactions deviate from canonical binding. While in a canonical RRM the $U_5$ base would be expected to stack onto RNP2 Y102, here $U_5$ rather stacks on $A_4$, as evidenced by the strong NOEs between $A_4$ H1′ and $U_5$ H5 and H6 resonances, as well as weaker NOEs between $A_4$ H8 and the $U_5$ H6 resonance (Supplementary Data 2). The ribose of $U_5$ is packed against Y102, F100, and M90 (Fig. 4C), and its base functional groups are involved in hydrogen-bonding with the conserved interdomain linker backbone and sidechains (e.g., the S134 hydroxyl, T135 backbone amide, and K137 NH3, Fig. 4B, D) and with the Y102 hydroxyl. Although some part of this H-bonding network would be lost, e.g., the contact to K137, a cytidine might also be accepted in this position, using an alternative hydrogen-bonding partner. Therefore, this contact is likely not fully sequence-specific.

The most surprising element of the structure was the interaction between a highly conserved binding pocket of RRM2 and the specific recognition of $U_6$. $U_6$ lies under RRM2 α-helix 2 with its ribose ring in contact with A193, M194, and K197 and its base carbonyl O4 hydrogen-bonded by both the NH3 of K196 and the sidechain HE1 of W215 (Fig. 4D). This double H-bond to O4 renders this contact fully sequence-specific. Contacts to the $U_6$ and $U_7$ phosphate groups by H189 and K197 sidechains, respectively, further stabilize the $U_6$-RRM2 interaction, as evidenced by a large number of intermolecular NOEs (Supplementary Fig. 4A). These interactions with RRM2 α-helix 2, holding the $U_6$ nucleotide like a claw, allow for an unusual reversal of the RNA backbone direction at $U_6$, which is rotated by 120 degrees around a vertical axis compared to $U_5$ (see Fig. 4D). The contacts to the $U_6$ phosphate group by R88 and Q39 of RRM1 (Fig. 4E) help position two RRMs relative to each other, which explains their cooperative binding since both RRMs contribute to $U_6$ binding.

The $U_7$ phosphate is fixed by a salt bridge to K197 on RRM2, while the sidechain NH2 of N37 on the tip of the N-terminal eRRM1 hairpin extension interacts with the $G_8$ phosphate (Fig. 4D). The $U_7$ base is not

sequence-specifically recognized. Finally, the $G_8$ nucleotide is not well-defined. Overall, in this conformation, the phosphate groups from $U_3$ to $G_8$ are hydrogen-bonded to one or two protein sidechains, with some originating from the RRM1 extension (N37 on β-strand −1 and Q39 on β-strand 0). Altogether this structure suggests recognition of a $N_2Y_3A_4Y_5U_6N_7N_8$ consensus sequence (Y is a pyrimidine) by the tandem RRMs of DND1 as cytidines in the positions of $U_3$ and $U_5$ would likely be accepted without significant disruption of the hydrogen-bonding network.

The binding topology on the eRRM1 and RRM2 is unusual with the involvement of a nucleotide-binding pocket in α-helix 2 of an RRM. As the β-hairpin extension and RNA-binding residues on eRRM1 are conserved in the hnRNPR-like family of RNA-binding proteins (Supplementary Fig. 1A, D), it seems likely that the RNA-binding mode of the eRRM1 is conserved. Although the interdomain linker, comprising amino acid residues TEK, is unique for DND1 within this family, it is possible that the linkers of the other family members might bind RNA in a similar fashion using an alternative hydrogen-bonding network. Notably, DND1 is the only family member with a noncanonical β-sheet binding surface for RRM2, lacking aromatic residues.

## Mutation of key RNA-binding residues compromises the RNA interaction in vitro

To further understand the structure, we replaced several RNA interacting residues of the tandem RRMs of DND1 with alanine. We then tested the effect of these substitutions on RNA binding by NMR titrations, after confirming that the protein fold was not affected (Supplementary Fig. 5A). The mutant of a canonical RRM1 RNP residue R98A failed to bind CUUAUUUG (Supplementary Fig. 5B), confirming in vitro that the eRRM1 is the primary RNA-binding surface of DND1, consistent with our RNA-IP and qPCR experiments (Fig. 1B and Supplementary Fig. 2). Of note, other RRM1 RNP mutants could not be tested. Although we could express mutant F61A, it precipitated during the purification. Y102A could not be expressed in the soluble fraction at all, despite the fact that its equivalents have previously been used in several studies as RNA-binding mutants[9,24–26]. Although its solubility in

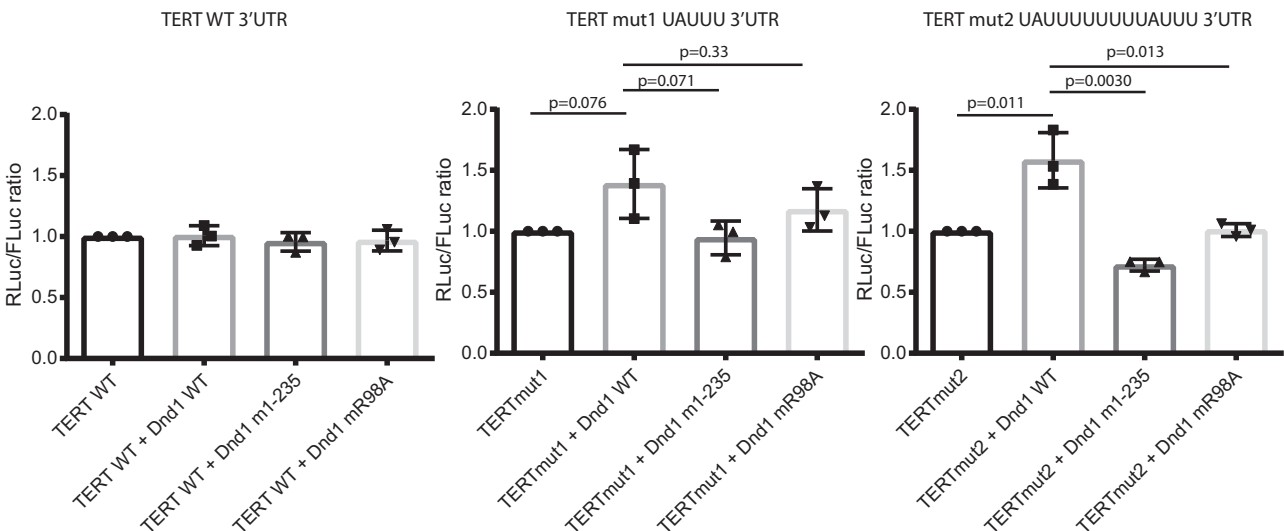

**Fig. 5 | Introduction of an AU-rich motif into a UTR is sufficient, RNA binding by the tandem RRMs is not enough for target rescue by DND1.** A wild-type TERT-UTR psiCHECK2 luciferase construct or the same construct with the introduction of a single UAUUU or double UAUUUUUUUAUUU DND1 tandem RRM target site, and either wild-type or mutant FLAG-tagged DND1 were transfected into HEK293T.

Relative luciferase activity is the ratio between Renilla control and firefly luciferases, adjusted to 1 for 100%. The results are represented as means and SD from three independent experiments. *P* values from two-tailed Welch's *t*-test. Immunostaining with anti-FLAG antibody in Supplementary Fig. 2 shows that DND1 and all mutants are well expressed in HEK293T cells. See also Supplementary Fig. 6.

cellulo and in vivo might be improved, its inactivity could result from an overall instability of the protein. Mutation of M90 on β2 of eRRM1, a residue interacting with $U_5$, also abolishes RNA binding (Supplementary Fig. 5B). Although the linker T135 sidechain makes several specific contacts with $A_4$ or $U_5$, mutation of this residue to A does not reduce RNA binding. Most likely, other residues in the linker could compensate using an alternative hydrogen-bonding network for binding to $U_5$. K197A and W215F mutations of residues of the unusual RNA-binding pocket of RRM2, resulted in weaker binding to RNA. Smaller chemical shift changes were seen in W215F compared to the WT upon RNA titration. In the K197A mutant, the NMR signals of the complete RRM2 α-helix 2 were exchange-broadened in the RNA-bound state, indicating a less stable RNA binding. We performed ITC experiments (Supplementary Fig. 5C) to quantify the loss of RNA binding and obtained Kd values of 2.6 μM for K197A and >10 μM for W215F, compared to 0.7 μM for WT DND1–RRM12 binding to CUUAUUUG (Fig. 2A). These were significantly less than the Kd for RRM1 alone (>54 μM, Fig. 2C), suggesting that the single mutations do not fully destroy the RRM2-binding pocket. These assays confirmed that the eRRM1 RNP is essential for RNA binding and that the atypical RRM2-binding pocket stabilizes the primary eRRM1–RNA interaction.

**Introduction of an AU-rich motif into a reporter gene 3′UTR is necessary and sufficient for target repression by DND1**

To investigate how the reduced RNA binding caused by these mutations affects the function of DND1 in cellulo, we tested these single amino acid mutants in the context of the full-length protein in a luciferase-based gene reporter assay. Kedde et al. used such assays to show that DND1 protects certain mRNA targets from miRNA-mediated repression[9]. Thus, we transfected a psiCHECK2 dual luciferase plasmid bearing a partial p27/CDKN1B 3′UTR sequence into HEK293T cells. The partial UTR, cloned downstream from the Renilla luciferase ORF, contains two target sequences for miR-221 and two putative DND1 binding sites. This stretch of sequence was identical to that shown in ref. 9. to be sufficient for the protection of the target by DND1. Co-transfection of the reporter plasmid with a miR-221-3p mimic (miR-IDIAN, Dharmacon) produced the expected decrease of luciferase activity compared to that observed with co-transfection of a negative control scrambled miRNA mimic (Supplementary Fig. 6). Co-

transfection of a plasmid expressing FLAG-tagged wild-type DND1 with the negative control miRNA increased luciferase expression from the reporter, demonstrating the functional effect of DND1 on this target. In contrast, FLAG-tagged wild-type DND1 showed only a mild, non-statistically significant effect on the repression of luciferase by miR-221. We concluded that DND1 was unable to counter the suppression of p27/CDKN1B 3′UTR by miR-221, at least under the conditions tested here. The effect of the R98A and 1–235 dsRBD-truncation mutants did not differ significantly from that of WT DND1. We reasoned that possibly DND1 was not able to displace the miRNA that was present in abundance from the beginning of the experiment and possibly tightly bound to the target mRNA prior to a sufficiently high expression of the protein. The data suggested that this assay system was not suited to investigate the functional role of DND1–RNA binding on miRNA regulation. Therefore, we focused efforts on the inductive effect of DND1 alone. We inserted the full 3′UTR from the telomerase reversed transcriptase gene (TERT) into the psiCHECK2 dual luciferase plasmid. This mRNA was not expected to be targeted by DND1 as it lacks AU-rich regions and is not present in recent PAR-CLIP and DO-RIP DND1 target datasets[7,11]. Upon transfection of this reporter into HEK293T cells together with expression plasmids for either wild-type or mutant full-length FLAG-tagged DND1, we do not observe any effect on luciferase activity (Fig. 5). Yet, insertion of a single UAUUU (the central pentanucleotide bound to the tandem RRMs in our structure) into the TERT 3′UTR was sufficient to increase luciferase activity upon transfection of wild-type DND1. This increase became statistically significant if two consecutive UAUUU sequences were introduced, spaced by a UUUU tetranucleotide, with the rationale to prevent steric hindrance in case of binding of two copies of DND1, but short enough to facilitate an avidity effect in case of binding of only one copy. Transfection of the R98A eRRM1 RNP mutant or the truncation mutant lacking the dsRBD (DND1 1–235) did not increase luciferase activity. These results indicate that the presence of an AU-rich motif in the 3′UTR is necessary and sufficient for DND1-mediated target regulation. While the dsRBD-truncation mutant was unable to raise target expression here, our RNA-IP followed by q-RT-PCR (Fig. 1B and Supplementary Fig. 2) showed that it still is able to bind RNA in cellulo. Therefore, RNP-mediated motif recognition by the tandem RRMs of DND1 appears to be necessary, but not sufficient for target regulation.

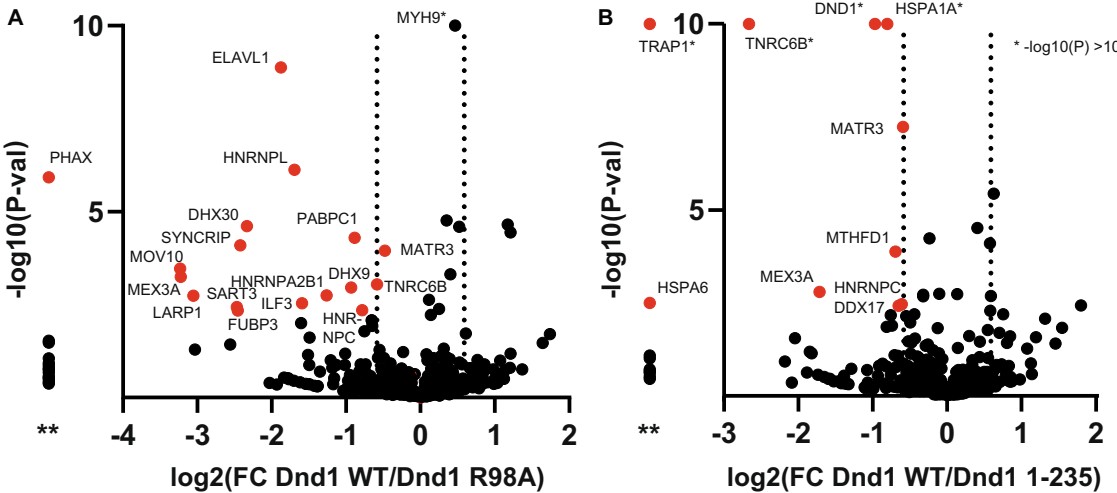

**Fig. 6 | Proteomics analysis revealed differential interaction networks of targets of mutant versus WT DND1.** In total, 769 proteins were quantified and used for *t*-testing to generate the volcano plot indicating the log₂ fold change (FC Dnd1 WT/Dnd1mut) ratio on the *x* axis and −log*P* value on the *y* axis for each protein. * Shows proteins with −log(*P* val) > 10. **A** 17 proteins (in red) were present at at least 1.5-FC lower levels in the R98A RNA-binding mutant as compared to DND1 WT. **B** Nine proteins (in red) were present at at least 1.5-fold change lower levels in the 1–235 dsRBD-truncation mutant as compared to DND1 WT. Multiple two-sided *t*-tests were performed using FDR = 10%, see Supplementary Data 1. ** In both panels, the column left of the *y* axis shows proteins present in WT but completely missing in the mutants. Source data is provided in columns G, H, O, and P in the *t*-tests sheet of Supplementary Data 1, raw data are deposited at ProteomeXchange (see "Data availability").

This suggests that the dsRBD might have a downstream role in DND1-mediated gene regulation that is apparently independent of RNA binding.

## The interactome of DND1 targets in HEK293T is enriched for mRNA stabilizing proteins and proteins present in RNP granules

To learn more about the context of the regulatory mechanism of DND1 binding, we transfected HEK293T cells with FLAG-tagged WT DND1, the R98A RRM12 RNA-binding mutant, and the 1–235 dsRBD-truncation mutant. We then performed immunoprecipitation and Liquid Chromatography Electrospray Ionization Tandem Mass Spectrometric (LC/ESI-MS/MS) analysis on three biological replicates, for identification and quantification of the pulled-down interactome (Fig. 6 and Supplementary Data 1). In total, overall mutants and replicates, 769 proteins were identified, of which 25 and 20 were found to be present at statistically different levels for the R98A mutant and the 1–235 dsRBD-truncation mutant, respectively, as compared to the interactome of WT DND1 targets (Supplementary Data 1, two-sided *t*-tests using FDR = 10%). Pleasingly, 29 of these 45 differentially present proteins were also found in ref. 11. in the DND1 interactome from HEK293T cells. Notably, however, our dataset (including all proteins equally pulled down by WT DND1 and mutants) did not include members of the CCR4-NOT complex, which deadenylates mRNAs as the first step in their degradation. This may have been due to the different methods of DND1 overexpression in the two protocols: transient expression vs. a stable cell line, as well as different lysis and wash conditions. Seventeen proteins were present at statistically significant lower levels with at least 1.5-fold change (FC) in the pulldown with the R98A RNA-binding mutant (see Fig. 6A and Supplementary Data 1). This set included other RNA-binding proteins that bind U-rich and AU-rich RNA (ELAVL1/HuR, HNRNPC). Gene Ontology analysis using the EnrichR tool[27] (Supplementary Fig. 7) showed that this list was enriched for mRNA stabilizing proteins (HNRNPC, ELAVL1/HuR, DHX9, SYNCRIP, LARP1, and PABPC1). Moreover, the list was also enriched for "RNP granules as a cellular component" (due to PABPC1, LARP1, MOV10, DHX9, HNRNPL, DHX30). We validated ELAVL1, that protein present at the most statistically significant reduced level in the sample of the R98A RNA-binding mutant, using western blot (Supplementary Fig. 8A) and looked for co-localization with DND1 using

immunofluorescence microscopy. FLAG-tagged DND1 shows diffused cytoplasmic- and nuclear localization in HEK293T, similar for WT and the R98A RNA-binding mutant, while the 1-235 dsRBD-truncation mutant is diffusely expressed in the nucleoplasm and exhibits a punctate staining in the cytoplasm (Supplementary Fig. 8B). As expected, ELAV/HuR is located mainly in the nucleus. It is known to shuttle between the nucleus and cytoplasm[28], where it stabilizes mRNAs by binding to AU-rich elements (AREs)[29,30]. One of its many functions is the regulation of mESC differentiation[31]. Just like DND1, it also has been reported to increase mRNA stability by blocking microRNA-targeting[32]. The RNA helicase MOV10, also enriched in our DND1 WT immunoprecipitations over the R98A RNA-binding mutant, is expressed in the cytoplasm and exhibits diffused staining there similar to WT and R98A DND1 (Supplementary Fig. 8C). Both ELAVL1/HuR and MOV10 are present in stress granules[33] and are enriched in P-bodies[34], two important classes of cytoplasmic RNP granules. Cytoplasmic RNP granules are hotspots for post-transcriptional gene regulation, which depend upon complex networks of protein−RNA interactions, low-complexity protein sequences, and liquid−liquid phase separation (LLPS) for their formation[35]. In addition, several other proteins enriched in the WT DND1 target HEK293T interactome are also enriched in stress granules and/or P-bodies (Supplementary Data 1)[33,34]. Such enrichment in cytoplasmic granules was not observed for either ectopically expressed WT or mutant DND1 proteins nor for endogenous ELAV or MOV10 in transiently transfected HEK293T cells (Supplementary Fig. 8B, C).

Nine proteins were present at statistically significant lower levels of at least 1.5FC in the pulldown of the 1–235 truncation mutant. This set was still GO-enriched in RNA-binding terms (EIF4B, HNRNPC, MEX3A, DDX17, MATR3, TNRC6B). This suggested that the dsRBD might contribute to the binding of a subset of RNA targets regulated by these RBPs, or it recruits them to modulate the target specificity of DND1. Interestingly, TNRC6B is known to recruit CCR4-NOT. Finally, a surprising finding from the analysis by GO-term was enrichment of the "ATPase activity" classification. DDX17 is an RNA helicase, while HSPA1A, HSPA6, and TRAP1 are chaperones belonging to the heat shock protein family that assists protein folding upon ATP hydrolysis; these proteins recognize hydrophobic surfaces of unfolded proteins and partially folded intermediates, thereby preventing protein

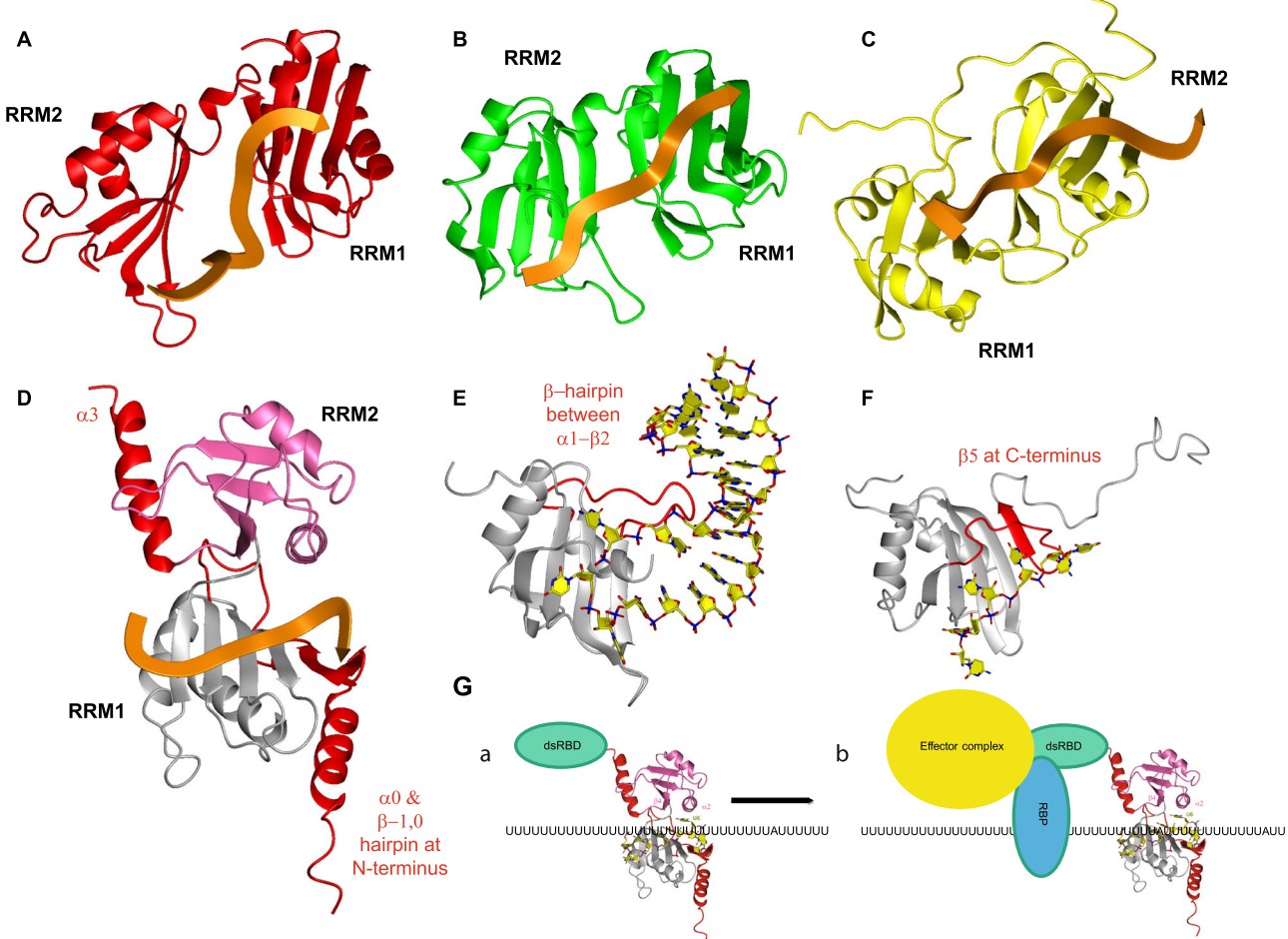

**Fig. 7 | Structural comparison of the DND1 tandem RRM-RNA complex to five other RRM-RNA complexes and model of DND1 action mechanism. A–D** Tandem RRM-RNA complexes. **D–F** RRMs using structural extensions to the canonical RRM fold to increase affinity to their RNA targets. **A** The N-terminal two RRMs of HuD bound to AU-rich RNA. **B** The N-terminal two RRMs of PABP bound to poly(A) RNA. **C** The RRMs of TDP-43 bound to UG-rich RNA. **D** The RRMs of DND1 bound to AU-rich RNA with extensions to the canonical RRM fold shown in red. **E** FUS RRM bound to a stem-loop from hnRNPA2/B1 pre-mRNA. **F** RRM2 of PTB bound to CU-rich RNA. **G** model of AU-rich mRNA target regulation by DND1.

aggregation[36]. This suggested that even in the HEK293T cellular environment, there might have been folding issues with the DND1 dsRBD, a domain we were unable to obtain in soluble form for structural studies due to persistent aggregation during recombinant protein production. The ATPase activity of co-purified folding chaperones could explain the remarkable assignment of enzymatic activity to the dsRBD[37,38].

## Discussion

### Unique arrangement of DND1's tandem RRMs dictates the conformation, orientation, and accessibility of bound RNA

We determined the solution structure of the extended tandem RRMs of DND1 complexed with AU-rich RNA, which clarified the mechanism of target RNA recognition by this RNA-binding protein that is essential for germ cell survival in higher vertebrates. Previously solved tandem RRM-RNA complexes have shown that the two canonical β-sheet surfaces accommodating the RNA either form a cleft, like Sex-Lethal (Sxl)[39] and Human Antigen D (HuD—Fig. 7A)[22], or an extended surface (Poly-A Binding Protein (PABP—Fig. 7B)[18] and TAR DNA-binding protein 43 (TDP-43—Fig. 7C)[20]). In all these structures, RNA stretches of 8–10 nucleotides are bound in a canonical fashion, with higher affinity and specificity than if a single domain is used. In all aforementioned tandem RRM structures, the bound RNA adopts an extended conformation. The RRM-RRM orientation and RNA-binding topology are rather different in

our DND1–RNA structure, which is a result of four structural features found so far only in this complex: the lack of a canonical RNA-binding surface for RRM2, an ultra-short inter-RRM linker, an extended RNA-binding surface of eRRM1, and the presence of an RNA-binding pocket on α-helix 2 of RRM2. The complex embeds only a short canonical RNA-binding stretch ($U_3$-$A_4$), which is followed by binding of $U_5$-$U_7$ in a highly unusual manner. Indeed, $U_6$ is bound by RRM2 α-helix 2, resulting in a 120° rotation of $U_6$ compared to $A_4$ and the $U_6$ phosphate. $U_7$ ribose and G8 phosphate are bound by the tip of the eRRM1 extension. Binding by the linker residues supports the RNA in this unique conformation, its short length being likely crucial to bridge nucleotides specifically recognized by RRM2 and eRRM1. The path of the RNA backbone is reversed, and the RNA is more compacted than in previously determined tandem RRM-RNA complexes; the $U_3$-$G_8$ phosphates span ~21–23 Å (Fig. 7D), while for example, in PABP an equivalent stretch of six nucleotides spans ~26–28 Å (Fig. 7B). Such backbone reversal capacity might help to fold the RNA, or alternatively, the tandem RRMs might be suited to recognize an RNA that is in an extended stem-loop conformation. Also, the central RNA residues are not solvent-accessible compared to other tandem RRM-RNA complexes. This structural feature would be consistent with DND1 possibly acting as a steric inhibitor of effector complexes targeting proximal binding sites, as suggested for the miRNA seed-complementary sequences targeted by miRISC in p27/CDKN1B and LATS2[9].

## Structural extension of the eRRM1 increases RNA-binding affinity and stabilizes a backbone turn in the recognized RNA

While several extensions to the canonical RRM fold have been described, either extending the β-sheet surface by one or several strands or adding an α-helix at the N or C-terminus[12], the DND1 N-terminal extension of a β-hairpin packed on a third α-helix is so far restricted to the hnRNPR-like family of proteins (Supplementary Fig. 1A, D). An X-ray structure of a similar eRRM from another member of this family, SYNCRIP/hnRNPQ, in its free form has been solved[23]. The latter is highly similar to the DND1 eRRM1 with the exception of the β3′/3″ hairpin and a small shift in the orientation of the N-terminal extension. These differences are likely due to RNA binding in our structure or, alternatively, the presence of an additional N-terminal acidic domain (AcD) found in the SYNCRIP structure. Our structure reveals that this N-terminal extension is essential for increasing the affinity to the RNA by fixing the backbone of $U_6$ and $G_8$ on the eRRM1 via Q39 and N37, respectively (Fig. 4D, E). Therefore, it is crucial for stabilizing the turn in the backbone observed in our complex. This is reminiscent of other extensions found in the RRM contributing to RNA binding like the β-hairpin found in FUS RRM[40] (Fig. 7E) and the fifth β-strand of PTB RRM2[41] (Fig. 7F).

## The hrRRM2 presents an atypical RNA-binding pocket, and its integrity is necessary for DND1 function

We have shown that the primary RNA interaction interface of DND1 lies on eRRM1. It is the proximity of the second RRM, lacking a canonical RNA-binding interface, but presenting an atypical pocket for stabilization of this primary binding, that makes the RNA-binding topology by the tandem RRMs unusual. Structures of several types of RRM domains without aromatic residues on the β-sheet surface have been described[12]. The qRRMs of hnRNPF use their β1/α1, β2/β3, and α2/β4 loops for recognition of G-rich sequences[42], while the ΨRRM of SRSF1 uses a conserved motif in the α-helix1 for purine-rich RNA binding[43]. However, our structure is the first example of an RRM mediating RNA binding via α-helix 2. We propose to call an RRM using this interface for RNA binding the hrRRM, for hnRNPR-like family-related RRM. We demonstrated the importance of the binding pocket on RRM2 by mutational analysis using in vitro binding assays and ITC (Supplementary Fig. 5B, C). It is also supported by its almost full conservation, not only in DND1 (Supplementary Fig. 1C) but also other members of the hnRNPR-family (Supplementary Fig. 1D). Thus, its RNA-binding mode is likely to be conserved. Our RRM2 structure is highly similar to that of the free RRM2 of RBM46 (RNA-binding motif protein 46, PDB: 2DIS)[44], including the orientation of the residues in the binding pocket. The importance of this pocket for DND1 function was demonstrated in functional studies in zebrafish, where the equivalent to the K197 mutant (K200T) was the only mutant outside of RRM1 causing loss of function[26]. Most other loss-of-function mutants in this study can be explained using our structure. We have already described that the zebrafish Y104 RNP mutant (the equivalent of Y102 in humans) is unstable in vitro. Even if it would be stable in vivo, interactions with $U_5$ would be lost. The equivalents of Y72, F89, and H121 in zebrafish dnd1 are Y70, F87, and H119 in human DND1, which are important residues stabilizing the RRM fold. Y70 is particularly important for the interaction between α-helices 0 and 1 in eRRM1, linking the core RRM fold and the N-terminal eRRM extension. Mutation of these residues most likely disrupts the fold of eRRM1. The only loss-of-function mutant that is not that easily explained is N94K, a mutant of the equivalent T92 in the human protein, situated in the β2–β3 loop. This residue is in close proximity to $G_8$ in our structure, but not well enough defined to interpret a specific interaction with the RNA. In the context of a longer RNA it could be involved in such specific binding. Finally, it should be mentioned that the Ter mutation, causing germ cell loss and testicular teratomas in mice[8], is a truncation at the equivalent of R190 in α-helix 2, further confirming that RRM2 and the dsRBD are essential domains

for DND1 function. The atypical binding pocket in RRM2 increases affinity and specificity to the readout of eRRM1 and creates a remarkable turn in the RNA backbone.

## Limited sequence specificity leads to the plasticity of RNA recognition

The RNA recognition observed in our structure unifies previously published seemingly contradictory data concerning the RNA recognition elements found enriched in DND1 targets. In fact, a combination of a UUA/UUU triplet as enriched in CLIP[11] was used in our structure determination as the RNA target. The motif $Y_3A_4Y_5U_6N_7N_8$ derived from our structure also fits with the [A/G/U]AU[C/G/U]A[A/U] motif enriched in RIP targets[7]. Moreover, this motif may be interpreted as a repetition of the UAU motif, containing 2 adenosines that are specifically recognized in our structure. We have not tested binding to an oligonucleotide containing two spaced adenosines, but the increased affinity (avidity) of RBPs binding to repetitions of high-affinity motifs has been demonstrated for several other RRM-containing proteins: hnRNPG[45], PTB[41], hnRNP C[46], and more recently ELAVL1/HuR[47]. Our structure also provides some insight into how the residues outside of the YAY motif could be recognized. For example, the binding pocket on RRM2 specifically recognizing $U_6$ in our structure (Fig. 4D) could not accommodate a cytidine without losing the double hydrogen bond to O4. Overall, it appears that the tandem RRMs of DND1 demonstrate a certain plasticity for RNA recognition where a range of sequences can be bound, but a $Y_3A_4Y_5U_6$ is necessary for high-affinity binding. Such high-affinity binding could be a prerequisite for the activation of downstream processes such as the recruitment of regulatory factors[48]. Here, we propose that the tandem RRMs bind RNA in a two-step mechanism. In a first step, a range of sequences may be preselected by low-affinity binding in order to attach DND1 to scan the 3′UTR (Fig. 7G, panel a). Upon encountering a high-affinity YAYU element, DND1 pauses at the central adenosine ($A_4$), while RRM2 locks the uridine ($U_6$) in its hrRRM binding pocket which can then initiate downstream processes from a fixed position (Fig. 7G, panel b).

## Role of the dsRBD

We have shown that DND1's tandem RRMs, like the majority of RRMs, are relatively sequence-tolerant[12]. On the other hand, we know that linear sequence motifs are often insufficient to fully capture RBP binding specificities[49]. Specificity may be increased due to contextual features, either in the form of bipartite motifs, such as recently found for FUS[40], preference for a nucleotide composition flanking a high-affinity linear motif, or due to the favoring of specific structural motifs adjacent to the linear motif. While the RNA-binding surfaces of the tandem RRMs are highly conserved within the hnRNPR-like family of proteins, the sequences of the dsRBDs of DND1, RBM46 and ACF1 (APOBEC1 complementation factor) are rather less conserved (Supplementary Fig. 1E). Thus, the highly specialized function of DND in germline development might originate from this domain. We have shown that the dsRBD is required for target regulation, either through direct or indirect recruitment of effector proteins, or maybe even by simply displacing other factors, but also that the dsRBD is not essential for the binding of two highly abundant RNA targets while contributing to the binding of a third one (Fig. 1B). While the dsRBD of DND1 lacks some canonical RNA recognition residues[50] (Supplementary Fig. 1E), this suggests that a noncanonical RNA-binding surface contributes to the binding of a subset of targets like SRSF2. The dsRBD might increase the target specificity of DND1 by recognizing a stem-loop motif adjacent to the linear motif recognized by the tandem RRMs. While we know that 3′UTRs are highly structured not only in vitro but also in vivo[51], it remains to be determined whether the 3′UTRs in the vicinity of the linear motifs targeted by DND1 are indeed enriched in secondary structure. Further structural studies should be undertaken to confirm that such structures can indeed be recognized by the full-length DND1.

Another possibility for increased target specificity of DND1 is cooperation with a binding partner. This was reported for NANOS2[10], which was shown to interact with CCR4-NOT[52,53], or other germline-specific RNA-binding proteins.

### DND1 is a germline-specific AU-rich element binding protein

DND1 binds UAUU which is contained in AU-rich sequence elements (AREs) in 3'UTRs that have long been known to target mRNAs for rapid degradation in mammals. AREs are divided into three classes with respect to the copy number of the canonical pentamer AUUUA sequence: several copies are dispersed between U-rich regions in class I, are clustered in class II, while class III are predominantly U-rich and lack these canonical pentamers[54]. More than 20 AU-rich RNA-binding proteins (AUBPs) have been identified, that control the fate of ARE-containing mRNAs[55]. Because DND1 CLIP and RIP-enriched targets do not necessarily contain the canonical ARE AUUUA pentamer target sequence, DND1 can be classified as a germline-specific AU-rich RBP (AUBP) targeting class III AREs. The recruitment of degradation machineries to mRNAs for their destruction is a unifying mechanism between several AUBPs[56–58], which could be shared by DND1 in one of its dual roles. The opposing, stabilizing role might depend on a network of other stabilizing RNA-binding proteins such as ELAVL1/HuR that is present in our pulldown. Interestingly, HuR shares with DND1 a possible function in stabilizing mRNA targets by inhibiting miRNA activity. Precisely how DND1 functions is likely target- and context-dependent as it may depend on the presence and concentration of competing factors. As multiple AUBPs may modulate the stability and translation of a single ARE-mRNA, questions of functional redundancy and additivity or antagonism arise. It is likely that variations in the relative amounts of mRNAs, in the AUBPs present within a certain cell type or developmental stage and in the binding affinities, determine both the identity of the targeted mRNAs and their fate[48]. On top of that, binding- and unbinding equilibria are altered in RNP-granular environments, where RNAs and RBPs are concentrated in a small volume. The sole fact that DND1 is specifically expressed in the germline will be a major contributing factor to its target specificity. Because of this highly complex context- and target dependence, the composition of the DND1 interactome found in our pulldowns cannot be confidently extrapolated to the process in developing germ cells. Thus, the interpretation of DND1 function based on reporter assays in non-native cell types should be addressed with caution. Similarly, the relevance of recognition motif derivation in large-scale DND1–RNA interaction studies performed in non-native cell types with transcriptomes differing from developing germ cells and using cross-linking that might overrepresent low-affinity motifs should also be interpreted with caution. The structural and biophysical work in this study raises our understanding of the requirements for a high-affinity RNA-binding motif in DND1. In turn, this helps to reinterpret the results from previous studies in order to better understand the complex gene-regulation networks during germline development and maintenance.

We have demonstrated here that DND1 prefers AU-rich over U-rich RNAs and that a central adenosine is required for high-affinity binding. The adenosine is specifically recognized by the eRRM1 binding pocket involving RNP2 and the interdomain linker (position "$N_1$"). This contrasts with the RRM3 of another AUBP, HuR, that recognizes both AU-rich and U-rich sequences with similar affinities, the latter slightly higher due to an avidity effect[47]. Adenosines are bound by HuR RRM3 in two different positions: either on the periphery, or β-strand 3 using RNP1 (position "$N_2$"). They are important to localize the protein at a precise position within the 3'UTR. Such a "locking" mechanism has been also proposed for the cytoplasmic polyadenylation element (CPE)-binding (CPEB) family of RNA-binding proteins. These RBPs bind the CPE sequence UUUUAU, which activates translation by cytoplasmic polyadenylation. The CPEB4 and CPEB1 tandem RRMs bind a UUUUA pentanucleotide sequence-specifically[17]. While the uridines are

bound by RRM1, the adenosine is specifically recognized by RRM2 using RNP2 (position "$N_1$"). RRM1 in isolation has a low affinity for U-rich RNAs and RRM2 does not bind U-rich sequences. Therefore, it is proposed that the protein is recruited to U-rich motifs through RRM1, after which it scans the 3'UTR in an open conformation until it encounters an adenosine in a consensus CPE and locks the protein onto this sequence. This is a similar mechanism as we now propose for DND1, although in our case the scanning for a high-affinity motif likely happens in a closed rather than open conformation since the isolated RRM1 does not bind U-rich sequences. This original mode of RNA target selection therefore could be a general mechanism for cytoplasmic RNA-binding proteins regulating RNA via their 3' UTR.

### A germ-granular environment for DND1?

Although we found that overexpressed DND1 likely is not localized to RNP granules in HEK293T cells, the presence of many proteins found in such condensates in its interactome raises the question if DND1-dependent target regulation could occur in an RNP granule-like environment in its native environment in developing germ cells. There is considerable overlap between stress, P-body, and germ granules (RNP granules unique to the germline), although the latter are thought to be maintained as distinct RNP assemblies in germ cells[59]. Interestingly, the folding chaperone TRAP1, highly enriched in WT over the 1-235 dsRBD-truncation mutant in our pulldowns, but also present in the HEK293T DND1 interactome published in ref. 11, is a mitochondrial isoform of cytosolic HSP90[60]. Germ-granular structures are often formed near mitochondria[59] and it has been shown that signaling lipids on mitochondrial membranes are responsible for the localization of components of intramitochondrial cement (IMC), a germ-granular structure in mouse spermatocytes[61]. The remarkable and reproducible presence of TRAP1 as its folding chaperone suggests that DND1 could be targeted to mitochondria in HEK293T and perhaps by a similar mechanism in developing germ cells[62]. Taken together with the other results from our differential interactome maps, this suggests DND1 might be part of germ granules localized near mitochondria regulating gene expression post-transcriptionally.

In conclusion, we provide here the first structural and mechanistic insight into the molecular mechanisms by which the RNA-binding protein DND1 regulates a subset of mRNAs and in turn might stabilize the fate of germ cells. Our results hint at a specialized function of the individual RNA-binding domains of DND1, where the tandem RRMs are mainly responsible for target binding and the dsRBD for binding of a subset of targets and target regulation. This possibly occurs through the recruitment of regulatory factors, other RBPs that modulate the specificity of DND1, or displacement of competing RBPs. Our structure unifies DND1–RNA recognition elements recently found enriched in genome-wide interaction studies and facilitates understanding of loss-of-function mutants previously described in the literature. In addition, we have demonstrated an additional means by which an RNA recognition motif can recognize RNA, extending the repertoire of this versatile and abundant RNA-binding domain.

## Methods

A table of resources and reagents is supplied as Supplementary Table 4. Further information and requests for resources and reagents should be directed to and will be fulfilled by the Lead Contact, Frédéric Allain (allain@bc.biol.ethz.ch).

### Protein expression and purification

DNA fragments encoding human Dnd1 RRM1 (12–139), RRM2 (136–227) or the tandem RRM12 (12–235) were PCR amplified from the source plasmid pCMV-SPORT6-hsDnd1, an IMAGE cDNA clone (clone ID MGC:34750; IMAGE: 5172595) purchased from Source BioScience (Nottingham UK) with the primers listed in Supplementary Table 1. They were cloned into the pET-M11 vector (EMBL) with an N-terminal

TEV-cleavable 6xHis-tag between the NcoI and Acc65I restriction sites, using BbsI instead of NcoI to cut the insert to circumvent insert-internal restriction sites. Protein mutants were obtained by PCR-based site-directed mutagenesis with the pET-M11-RRM12 (12–235) plasmid as a template according to the QuikChange protocol (Stratagene) and the primers listed in Supplementary Table 1. All protein constructs were expressed in *E. coli* BL21(DE3) cells (Novagen) in Studier-medium P-5052 supplemented with $^{15}NH_4Cl$ or P-50501 supplemented with $^{15}NH_4Cl$ and $^{13}C$-glycerol (CIL). Precultures were grown in PA-0.5G medium[63,64]. Random fractionally deuterated protein for the recording of triple-resonance spectra for backbone assignment was expressed in 100% $D_2O$ (CIL) in which the media components were directly dissolved. Protein was expressed for 60 h at 15 °C in the presence of 100 μg/mL Kanamycin. Cells were harvested by centrifugation at 4 °C, 15 min at 2600×$g$, and the cell pellet was resuspended in lysis buffer (20 mM Tris, pH 8, 1 M NaCl, 0.2% Triton-x-100 (w/v), 10 mM imidazole, and 2 mM 2-mercaptoethanol). Cells were lysed with two freeze-thaw cycles and three passes through the Emulsiflex cell cracker (Avestin). Before lysis 0.5 mg/ml lysozyme, 25 μg/ml DNAseI and 1 mM Pefabloc SC (Sigma-Aldrich) was added. After centrifugation at 4 °C for 20 min at 43,000 × $g$, the cleared supernatant was sterile-filtered and loaded onto 2 mL Ni-NTA beads (Qiagen), equilibrated with lysis buffer, per liter of bacterial culture. The column was washed with 10 column volumes of lysis buffer, 20 columns of lysis buffer without Triton and 5 column volumes of the same buffer with 30 mM Imidazole, before the protein was eluted with elution buffer (lysis buffer without Triton and with 330 mM imidazole). For cleavage of the $His_6$ tag, the pooled fractions were dialyzed against lysis buffer (1 M NaCl and no imidazole) in the presence of in-house purified TEV protease (1:100 w/w TEV:protein) at 4 °C overnight. Next day the TEV cleavage reaction was reloaded three times over a fresh Ni-NTA column to remove the $His_6$-TEV protease, the $His_6$-tag fusion and contaminating proteins. The proteins were concentrated with Vivaspin 20-mL centrifugal devices with 5000 or 10,000 MWCO (Sartorius) and buffer-exchanged into NMR buffer over PD-10 gel-filtration columns (GE Healthcare).

### RNA samples

Unlabeled RNA oligonucleotides were purchased from Dharmacon, deprotected according to the manufacturer's instructions, lyophilized, and resuspended twice in water for large-scale protein−RNA complex production, NMR buffer for titrations or ITC buffer. For the solid phase synthesis of selectively ribose-labeled oligos 2′-O-TOM protected ribonucleoside phosphoramidites and solid supports containing [$^{13}C_5$]-labeled ribose moieties were synthesized as described, followed by their sequence-specific introduction into the CUUAUUUG oligonucleotide[65].

### NMR sample preparation of protein−RNA complexes

Final protein was analyzed for nucleic acid contamination using (ed. Yeo, G.)$A_{260nm}/A_{280nm}$ and concentration was estimated using $A_{280nm}$ and a theoretical extinction coefficient of 18140 $M^{-1}cm^{-1}$ for RRM1, 5930 $M^{-1}cm^{-1}$ for RRM2, and 23470 $M^{-1}cm^{-1}$ for RRM12. RNA concentrations were estimated using OligoCalc[66]. In the final buffer exchange step the RRM constructs were added dropwise to a 10% molar excess of RNA in the presence of 10ul of SuperaseIn RNase inhibitor (Ambion) per sample, concentrated, and further purified by size exclusion chromatography on a Sephadex 75 10/30 column (GE Healthcare) in 100 mM $K_2HPO_4/KH_2PO_4$ pH 6.6, 1 mM DTT. The fractions containing the protein−RNA complex were concentrated to 400–700 uM with Vivaspin 5-mL centrifugal devices with 10,000 MWCO (Sartorius). Before the measurements, a 10% molar excess of RNA was added to saturate the protein as well as 10% v/v $D_2O$. Complexes were lyophilized before resuspending in $D_2O$ for NMR experiments that are conducted in deuterated solvent.

### Plasmids for cell culture assays

Total RNA was extracted from cultured human fibroblasts (GM03814, Coriell Institute for Medical Research, USA). In all, 1 μg was then used for reverse transcription reaction using Oligo(dT)$_{18}$ and M-MuLV Reverse Transcriptase RNaseH$^-$ (Finnzymes). The 3′UTR of TERT and fragments corresponding to positions 183−282 (according to Ensembl transcript ENST00000228872) of the 3′UTR of *p27*, including the predicted *miR-221* binding sites, were amplified from the cDNA templates using the appropriate primers in Supplementary Table 1 introducing XhoI and NotI restriction sites. The Dnd1 binding site was introduced into the TERT 3′UTR insert using PCR overlap extension using the primers in Supplementary Table 1. 3′UTR PCR products were directionally cloned downstream of the Renilla luciferase open reading frame (ORF) of the psiCHECK2 vector (Promega) that also contains a constitutively expressed firefly luciferase gene, which was used to normalize transfections. Dnd1 fragments encoding the full-length human protein (1−353) or a dsRBD truncation (1−235) were amplified as described for the protein expression plasmids and cloned BamHI/EcoRI into an in-house modified pcDNA3.1+ plasmid (Invitrogen) with an N-terminal FLAG tag cloned NheI/HindIII. All plasmids were confirmed by sequencing. Plasmids for transfections were prepared using the Nucleobond Xtra midiprep kit (Macherey-Nagel) according to the manufacturer's protocol.

### Transfections and dual luciferase activity analysis

HEK293T cells (Homo sapiens) were obtained from ATCC (CRL-3216). No further authentication of cell lines was performed. Cells were routinely checked for mycoplasma contamination. They were cultured in Dulbecco's Modified Eagle Medium (DMEM, Sigma) containing 10% fetal bovine serum (FBS, Sigma) including antibiotics (0.05 mg/mL of streptomycin, and 50 U/mL of penicillin (Sigma)) in a humidified incubator (Thermo Scientific Heraeus Series 6000 Incubators, Thermo Scientific) with 5% $CO_2$ at 37 °C.The cells were transfected with Lipofectamine 2000 Reagent (Invitrogen) after seeding them 16 h prior at 70,000 cells per well (24-well plate) or $2.8 × 10^6$ cells per 10 cm dish for immunoblotting analysis and RNA immunoprecipitation. For transfections in 24-well plates, Lipofectamine 2000 Reagent was diluted in serum-free medium (OptiMEM, GIBCO) and incubated for 5 min at room temperature. Plasmid DNA (0.5 μg per plasmid as indicated) and/or 50 nM final miR-221-3p miRNA mimic (miRIDIAN, Dharmacon) or control mimic was then added, vortexed, and incubated for 20 min at room temperature while cell culture media was exchanged to DMEM containing 10% FBS without antibiotics. Finally, the transfection complexes were added to the cell culture vessel in a dropwise manner while swirling. Transfection media were changed 6 h later to regular culture media. Luciferase activity was measured 48 h after transfection using the Dual-Glo Luciferase Assay System (E2920 Promega, USA) on a GloMax® Discover Multimode Microplate Reader (Promega, USA). The results are represented as means and standard deviation (SD) from three independent experiments.

### Immunoblotting analysis of protein expression and antibodies

Total cellular protein was extracted from $6 × 10^5$ HEK293T cells using a RIPA buffer (1% NP-40, 150 mM NaCl, 50 mM Tris-HCl pH 8.0, 0.5% sodium deoxycholate, 0.5% SDS) complemented with EDTA-free protease inhibitor cocktail (Roche) followed by brief sonication. Protein concentrations were determined by DC Assay (BioRad). For each sample, 14 μg of total cellular protein was separated on 12% SDS-PAGE gels and transferred on PVDF membranes. The following antibody was used: FLAG-M2-HRP (SIGMA, A8592). Immunoblots were developed using the Clarify ™ Western ECL substrate (BioRad) kit and were detected using an imaging system (ChemiDoc™ MP−BioRad). All membranes were stained using a coomassie blue staining solution to ensure equal loading. The analysis was performed in triplicate.

## RNA immunoprecipitation

The RNA-immunoprecipitation (RIP) procedure was adapted from Vogt and Taylor[67]. Briefly, subconfluent cells from one 10-cm dish were harvested 48 h after transfection, washed in PBS 1×, and cross-linked with 1% formaldehyde. Glycine (0.125 M final) was added to quench the formaldehyde. Cells were pelleted by centrifugation and washed with PBS 1×. Immunoprecipitation (IP) lysis buffer (50 mM HEPES at pH 7.5, 0.4 M NaCl, 1 mM ethylenediaminetetraacetic acid (EDTA), 1 mM DTT, 0.5% Triton X-100, 10% glycerol) containing 1 mM PMSF, protease inhibitors (Roche), and RNase inhibitor (Thermo Scientific) was added to the cell pellet. After sonication (Bioruptor, Diagenode), cell lysates were precleared with IP-Lysis buffer containing 1% BSA. 40 µl of magnetic FLAG-M2 beads (SIGMA, M8823) were added to the precleared cell lysate and incubated on a rotary wheel at 4 °C overnight. FLAG-M2 beads were washed with IP lysis buffer five times and pelleted by centrifugation. RIP buffer (50 mM HEPES at pH 7.5, 0.1 M NaCl, 5 mM EDTA, 10 mM DTT, 0.5% Triton X-100, 10% glycerol, 1% SDS) containing RNase inhibitor was added to the pellet and incubated 1 hour at 70 °C to reverse the cross-links. After centrifugation, the supernatant was used for RNA extraction using TRIzol reagent (Life Technologies) followed by a DNase-I treatment and a subsequent reverse transcription with oligo d[T]$_{18}$ using the GoScript RT kit (Promega). One-step RT-qPCR was performed using the SYBR FAST Mix optimized for Light-Cycler 480 (KAPA, KK4611) with primers listed in Supplementary Table 1. The results are presented as relative enrichment over the input ($2^{-\Delta Ct}$). $\Delta Ct$ is an average of (Ct [RIP] − (Ct [Input]) of technical triplicates with SD < 0.3. Three independent RIP experiments were performed.

## Immunoprecipitation and shotgun LC-MS/MS analysis of DND1 interactome

In all, $1.2 \times 10^6$ HEK293T cells were seeded the night prior to transfection per 10-cm dish. In total, 24 µg of DNA per sample was transfected using 50 µl of Lipofectamine 2000. Cells were harvested, cell pellet washed with 1×PBS and resuspended in 1 ml IP-Lysis buffer (50 mM Tris-HCl, pH 7.4, 150 mM NaCl, 1 mM EDTA, 0.5% Triton X-100) supplemented with PMSF and complete (Roche) protease inhibitors. Samples were incubated on ice for 10 min and sonicated. Lysates were centrifuged at $14,000 \times g$ at 4 °C. Protein was quantified using the DC protein assay (BioRad) and normalized to 1 mg/ml total protein. 40 µl of FLAG-M2 magnetic beads (SIGMA) (50% slurry) per IP were washed 3× using 750 µl IP-Lysis buffer. Lysate was added and incubated overnight at 4 °C.

For western blotting, after removal of the supernatant, the beads were washed gently twice in IP-Lysis buffer, followed by five times in 1× TBS (pH 7.4). Protein was eluted by adding 100 µl 0.1 M glycine pH 3.0 to the beads, followed by shaking for 6 min at 23 °C and 600 rpm. In total, 10 µl 0.5 M Tris-HCl, pH 7.4 was added to the supernatant and 10 µl of elution was used for WB analysis.

For shotgun LC-MS/MS analysis, the beads were washed twice with 100 µl 50% acetonitrile/NH$_4$CO$_3$ and once with 100 µl acetonitrile. In total, 45 µl of digestion buffer (10 mM Tris/2 mM CaCl$_2$, pH 8.2) and 5 µl trypsin (100 ng/µl in 10 mM HCl) were added. Digestion for 30 min at 60 °C was assisted by microwave. The supernatant was collected and peptides were extracted from the beads with 150 µl of 0.1% trifluoroacetic acid/50% acetonitrile. Supernatants were combined and dried, dissolved in 20 µl 0.1% formic acid, further diluted 1:10 in 0.1% formic acid of which 2 µl were injected.

Mass spectrometry analysis was performed on a nanoAcquity UPLC (Waters Inc.) connected to a Q Exactive mass spectrometer (Thermo Scientific) equipped with a Digital PicoView source (New Objective). Solvent composition at the two channels was 0.1% formic acid for channel A and 0.1% formic acid, 99.9% acetonitrile for channel B. For each sample, 2 µl were injected. Peptides were trapped on a Symmetry C18 trap column (5 µm, 180 µm × 20 mm, Waters Inc.) and

separated on a BEH300 C18 column (1.7 µm, 75 µm × 150 m, Waters Inc.) at a flow rate of 300 nL/min by a gradient from 5 to 35% B in 90 min, 60% B in 5 min and 80% B in 1 min. The mass spectrometer was operated in data-dependent mode (DDA), acquiring a full-scan MS spectrum (350–1500 m/z) at a resolution of 70,000 at 200 m/z after accumulation to a target value of 3,000,000, followed by HCD (higher-energy collision dissociation) fragmentation on the twelve most intense signals per cycle. HCD spectra were acquired at a resolution of 35,000 using a normalized collision energy of 25 and a maximum injection time of 120 ms. The automatic gain control (AGC) was set to 50,000 ions. Charge state screening was enabled and singly and unassigned charge states were rejected. Only precursors with intensity above 25000 were selected for MS/MS. Precursor masses previously selected for MS/MS measurement were excluded from further selection for 40 s, and the exclusion window was set at 10 ppm. The samples were acquired using internal lock mass calibration on m/z 371.1010 and 445.1200. Database searches were performed using the Mascot search engine against SwissProt. Scaffold (4.8.9) was used to filter sequence assignments with a protein threshold set to 1% false discovery rate (FDR), minimum number of peptides 2 and peptide threshold set to 0.1% FDR. Enrichment of pulled-down proteins was calculated by dividing the number of exclusive unique spectral counts found for that protein by the total number of exclusive unique spectral counts in that sample. To compare enrichment for WT DND1 to the mutants, multiple t-tests were performed on the median enrichment values of three independent pulldowns using GraphPad Prism 7.04. Discoveries were determined using the Two-stage linear step-up procedure of Benjamini, Krieger, and Yekutieli, with Q/FDR = 10%. Proteins with less than two exclusive unique spectral counts in the DND1 WT pulldown (median of the triplicate experiment) were excluded from the analysis (in red in Supplementary Data 1).

## Indirect immunofluorescence

HEK293T cells were transfected with Lipofectamine 2000 Reagent (Invitrogen) after seeding them 16 h prior at 200,000 cells per well (six-well plate). For transfections in a six-well plate, 2.0 µg of respective plasmid was used. The transfection protocol was as described for 24-well plates. Coverslips in a six-well plate were incubated with Poly-L-Lysine (Sigma P4707) diluted in PBS 1× to a final concentration of 1 µg/ml for 1 h at 37 °C, following this they were washed three times with PBS 1×. Twenty-four hours post transfection, cells were trypsinized and passaged onto Poly-L-Lysine coated coverslips to be processed for indirect immunofluorescence 48 h post transfection.

After aspirating media, cells were washed with PBS 1× and fixed with 3.7% formaldehyde solution for 10 min at room temperature, cells were washed with PBS 1× and permeabilized in ice-cold CSK buffer (0.1% Triton X-100, 10 mM PIPES, 100 mM NaCl, 3 mM MgCl$_2$, 300 mM Sucrose) for 7 min at room temperature. Cells were next blocked in PBS 1× supplemented with 1% BSA and 0.1% Tween-20 for 20 min at room temperature. Samples were incubated with primary antibodies diluted in blocking solution HuR (Proteintech 11910-1-AP; 1:250), MOV10 (Proteintech 10370-1-AP, 1:500), FLAG-M2 (Sigma F1804, 1:250) for 1 h at room temperature. Washes in PBS 1× + 0.1% Tween-20 were repeated three times and samples were incubated with secondary antibodies diluted in blocking solution (Alexa fluor 488 donkey anti-mouse IgG (IF 1:2000) Life Technologies A21202; Alexa fluor 546 donkey anti-rabbit IgG (IF 1:2000) Life Technologies A10040) for 1 h at room temperature. Samples were again washed three times with PBS 1× + 0.1% Tween-20 and incubated with DAPI at a final concentration of 0.1 µl/ml diluted in PBS 1× for 4 min at room temperature to counterstain nuclei. After a final wash in PBS 1×, coverslips were mounted on slides on a drop of antifade mounting media vecta shield (Vector labs H1000). Slides were imaged using a DeltaVision Multiplexed system with an Olympus IX71 inverse microscope equipped with a 60×1.4NA DIC Oil PlanApoN objective and a pco.edge 5.5 camera. Images were

deconvolved using softWoRx (Applied Biosystems) and further processed using FIJI.

## NMR data collection and assignments

All NMR spectra were recorded at 298 K on Bruker AVIII600 MHz, AVIII700 MHz, and Avance 900 MHz spectrometers equipped with cryoprobes and a Bruker AVIII750MHz spectrometer using standard NMR experiments if not mentioned otherwise[68]. The data were processed using Topspin 3.1 (Bruker) and NMRPipe[69] and analyzed with NMR-FAM-SPARKY[70]. Sequence-specific backbone assignments were 93% complete for non-proline residues and were obtained from 2D $^{1}$H-$^{15}$N HSQC, 2D $^{1}$H-$^{13}$C-HSQC, 3D $^{1}$H-$^{15}$N-NOESY ($t_{mix}$ = 120 ms) and a suite of 3D TROSY-based backbone experiments (HNCO, HN(CA)CO, HNCA, HN(CO)CA, HNCACB and HN(CO)CACB)[71,72] run on a random fractionally deuterated $^{13}$C,$^{15}$N-labeled (1:1.1) tandem RRM-CUUAUUUG complex. Sidechain protons were assigned to 80% completeness using 3D $^{1}$H-$^{15}$N-NOESY ($t_{mix}$ = 120 ms), 3D $^{1}$H-$^{13}$C-HMQC-NOESY ($t_{mix}$ = 70 ms), 3D $^{1}$H-$^{13}$C-HSQC-aromatic-NOESY ($t_{mix}$ = 80 ms), 3D (H)CCH-($t_{mix}$ = 21.7 ms) and HC(C)H-TOCSY ($t_{mix}$ = 23 ms) and 3D H(C)CH-COSY on a sample of fully protonated $^{13}$C,$^{15}$N-labeled (1:1.1) tandem RRM-CUUAUUUG complex. Sidechains in the free $^{13}$C,$^{15}$N-labeled RRM2 were assigned using H(C)(CCCO)NH-TOCSY ($t_{mix}$ = 17.75 ms) and (H)C(CCCO)NH-TOCSY ($t_{mix}$ = 17.75 ms) and transferred to the RRM12-CUUAUUUG complex where this was possible. RNA was assigned using the following set of spectra: 2D TOCSY ($t_{mix}$ = 60 ms), 2D NOESY ($t_{mix}$ = 150 ms) recorded on a 1:1 complex of unlabeled tandem RRM-CUUAUUUG complex. 2D $^{1}$H-$^{13}$C-HSQC recorded on 1:1 complexes between unlabeled tandem RRMs and selectively $^{13}$C-ribose-labeled C*UU*AU*UU*G or CU*UA*UU*UG* RNA where an asterisk after the nucleotide represents a $^{13}$C-labeled ribose moiety. 2D F$_2$ filtered NOESY ($t_{mix}$ = 120 ms) and 2D F$_1$-filtered, F$_2$-filtered NOESY recorded on $^{13}$C,$^{15}$N-labeled tandem RRMs in 1:1 complex with 1) unlabeled CUUAUUUG RNA 2) selectively $^{13}$C-ribose-labeled C*UU*AU*UU*G RNA 3) selectively $^{13}$C-ribose-labeled CU*UA*UU*UG* RNA. Sugar puckers in the complex were identified from 2D $^{1}$H-$^{1}$H- TOCSY ($t_{mix}$ = 60 ms). Strong H1′-H2′ and weak H3′-H4′ cross-peaks defined all puckers as C2′-endo. All χ dihedral angles were restrained to anti conformations based on lack of strong intraresidue H1′-H6/H8 NOEs. Intermolecular NOEs were identified using 2D $^{13}$C F$_2$ filtered 2D NOESY ($t_{mix}$ = 60 ms) and 3D F$_3$ filtered, F$_2$-edited $^{13}$C HMQC-NOESY ($t_{mix}$ = 70 ms) in D$_2$O. Intramolecular NOEs of RNA were identified using 2D $^{13}$C F$_1$ filtered, F$_2$ filtered NOESY ($t_{mix}$ = 150 ms) in D$_2$O.

## NMR titrations

NMR titrations were performed by adding unlabeled concentrated RNA (1–5 mM) to $^{15}$N-labeled protein (0.1–0.2 mM) in NMR buffer (20 mM MES pH 6.6, 100 mM NaCl) and monitored by $^{1}$H-$^{15}$N-HSQC. To monitor the chemical shift perturbations of the tandem RRM mutants upon addition of RNA, 1:1 complexes were directly prepared in NMR buffer (100 mM K$_2$HPO$_4$/KH$_2$PO$_4$ pH 6.6, 1 mM DTT) as described under "NMR sample preparation".

## NMR relaxation and RDC measurements

Backbone dynamics data of the tandem RRMs in complex with CUUAUUUG were recorded on a 1:1.1 complex of random fractionally deuterated $^{13}$C,$^{15}$N-labeled protein with unlabeled RNA on a Bruker AVANCE 750 MHz spectrometer at 298 K. The heteronuclear $^{15}$N–{$^{1}$H} NOE values were measured with reference and NOE experiments recorded in an interleaved fashion, employing water flip-back pulses. Heteronuclear NOE values are reported as the ratio of peak heights in paired spectra collected with and without an initial period (4 s) of proton saturation during the 5-s recycle delay. $^{15}$N T1 and T2 values were measured using TROSY-based pseudo-3D experiments employing flip-back pulses and gradient selection[73]. T1 spectra were acquired with delays, T = 40, 150, 300, 500, 900, 1500, 2200, and 3000 ms,

T2 spectra were acquired with CPMG delays, T = 17, 34, 51, 68, 103, 137, 188, and 239 ms. $^{15}$N T1 and T2 values were extracted by plotting the decay of HN volumes and fitting the curves with standard exponential equations using the nlinLS lineshape fitting program within the NMRPipe package[69]. Residual dipolar coupling (RDC) restraints were extracted using $^{1}$H-$^{15}$N TROSY run on a fully protonated $^{15}$N-labeled (1:1.1) tandem RRM-CUUAUUUG complex in NMR buffer (isotropic dataset) and NMR buffer mixed with 4.2% C12E5 polyethylene glycol/hexanol medium[74] (anisotropic dataset). RDCs were derived by subtracting the isotropic from anisotropic $^{1}$H chemical shift differences between TROSY and anti-TROSY spectra recorded in an interleaved manner. Only un-overlapped peaks were analyzed and RDC restraints were employed only for structured residues with $^{15}$N het-NOE values larger than 0.6. The RDC rhombicity and anisotropy components were determined in CYANA by grid-search using an initial protein structure and further refined in subsequent structure calculations.

## Structure calculation and refinement

Intramolecular protein distance restraints were derived from 3D $^{1}$H–$^{15}$N NOESY ($t_{mix}$ = 80 ms) and 3D $^{1}$H–$^{13}$C HMQC-NOESY ($t_{mix}$ = 70 ms), 3D $^{1}$H-$^{13}$C HSQC-aroNOESY ($t_{mix}$ = 80 ms) and 2D NOESY ($t_{mix}$ = 80 ms). The protein resonance assignments of the tandem RRM-CUUAUUUG complex and a list of manually assigned protein core NOEs were used as input for automatic peak picking and NOESY assignment using ATNOSCANDID[75,76] in a two-step procedure. First, intra-RRM NOEs were assigned by including only resonance assignments for one individual RRM in two separate runs. Second, an ATNOSCANDID NOE assignment was performed using all resonance assignments. In this run, a list of upper limit distance restraints combining the restraints obtained in the runs performed with assignments for the individual RRMs was included. This procedure was found to be necessary to obtain the correct global topology for the two RRMs. The resulting peak lists were then checked and supplemented manually with additional picked peaks and several critical manual NOE assignments. The optimized NOESY peak lists from this procedure were re-assigned with the NOEASSIGN module of CYANA 3.96[77] while iteratively adjusting and keeping key manual assignments fixed during iterative refinement of the structure. Intraprotein hydrogen bonds were identified for HN resonances which were protected during hydrogen–deuterium exchange by reference to intermediate structures and added as restraints in further rounds of structure calculation. Following the determination of the protein structure in the bound state, the structure of the complex was determined. Intra-RNA and intermolecular NOESY peaks were picked and assigned manually and calibrated using known distances of H5-H6 cross-peaks of pyrimidines. In structure calculations including the RNA, unambiguous intermolecular NOEs were included first for the initial positioning of the nucleotides. Intermolecular NOEs with ambiguous assignments were then included as ambiguous restraints in CYANA and assigned unambiguously based on preliminary calculations. To further confirm the intermolecular restraints, we back-calculated short intermolecular distances from our final structures and inspected the spectra for the completeness of intermolecular NOEs. Final structure calculations in CYANA included intraprotein, intra-RNA and intermolecular NOEs, protein dihedral backbone restraints, intraprotein hydrogen bond restraints, and restraints for sugar pucker and syn or anti conformations identified from NOE patterns of H6 or H8 resonances. Protein dihedral backbone restraints derived from TALOS+[78] and additional manually defined β-hairpin turn restraints were used for the N-terminal β-hairpin extension. In the final structure calculation, 500 structures were calculated with CYANA, and the 50 lowest energy structures were selected for refinement with the SANDER module of AMBER12[79] using the ff12SB force field with implicit solvent, and 20 were selected based on the criteria of lowest amber energy and lowest intermolecular restraint violations.

## Isothermal titration calorimetry

ITC experiments were performed on a VP-ITC microcalorimeter (Microcal). Freshly purified protein was buffer-exchanged by gel filtration into ITC buffer (100 mM $K_2HPO_4/KH_2PO_4$ pH 6.6, 1 mM 2-mercaptoethanol). RNA (100–400 μM) was dissolved in ITC buffer and titrated into protein (3.5–11 μM) in 2 μL followed by 8 μL (RRM12) or 10 μL (RRM1) steps every 300 s at 25 °C with a stirring rate of 307 rpm. Raw data were integrated, normalized for the molar concentration, and analyzed using the Origin 7.0 software according to a 1:1 RNA:protein ratio binding model. N was set to 1 for weak interactions.

## Reporting summary

Further information on research design is available in the Nature Research Reporting Summary linked to this article.

## Data availability

The coordinates for the structural models of DND1–RRM12:CUUAUUUG have been deposited in the Protein Data Bank under ID code PDB 7Q4L, and the assignments have been deposited at BMRB under ID code BMRB: 34675. The mass spectrometry proteomics data (Fig. 6) have been deposited to the ProteomeXchange Consortium[80] via the PRIDE[81] partner repository with the dataset identifier PXD024666. *MS/MS sample protein identification*: Database searches were performed using the Mascot search engine against SwissProt. Source data are provided with this paper.

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

## Acknowledgements

We are grateful to Gunter Stier (BZH, Universität Heidelberg) and Dr. Arie Geerlof (PEPF Helmholtz Zentrum München) for providing us with expression plasmids and protocols that were developed at EMBL Heidelberg. We thank Dr. Julien Boudet for assistance with ITC measurements, Dr. Antoine Clery for cloning the RRM2 expression plasmid and assistance with ITC, Dr. Fred Damberger for assistance with NMR experiments and structure calculations, Dr. Markus Blatter and Dr. Yaroslav Nikolaev with structure calculation and IT support respectively, Mauro Zimmermann for assistance with RNA preparation, Dr. Daniel Cirera-Salinas for cloning the psiCHECK2-TERT plasmid and assistance with luciferase assays. We thank Dr. Nina Ripin and Dr. Fred Damberger for critical reading of the manuscript. NMR was carried out at the Biomolecular NMR Spectroscopy Platform BNSP of the ETH Zurich. MS was carried out by the Functional Genomics Centre Zurich (FGCZ). This work was supported by the Promedica Foundation (to M.M.D.) and the Swiss National Science Fund SNF (120% Support Grant to M.M.D., SNF grants 31003A_149921 and 31003A_170130 to F.H.-T.A., 31003A_173120 to C.C.) and the NCCR RNA & Disease to F.H.-T.A., C.C., and J.H.

## Author contributions

M.M.D., F.H.-T.A., and C.C. designed the experiments and wrote the paper. M.M.D. prepared samples, solved the structure, and ran ITC experiments. H.W. performed luciferase, RIP, and qPCR assays. T.K. prepared mutants and samples. R.A. performed the fluorescence microscopy and assisted in the MS data analysis. F.E.L. designed and performed the RRM2 titration experiments. C.v.S. prepared mutants. U.P. synthesized selectively labeled RNA oligos. M.M.D., F.H.-T.A., H.W., and C.C. analyzed the data. F.H.-T.A., J.H., and C.C. provided funding and resources.

## Competing interests

The authors declare no competing interests.
