## [Peer Review File · Nature Communications]

REVIEWER COMMENTS

Reviewer #1 (Remarks to the Author):

Summary:

In this manuscript, the authors determine the solution structure of the tandem RRM domains of the dead end (DND1) protein bound to RNA. In addition, in cellulo evidence supports the structural findings. Key findings include an unprecedented mode of RNA recognition by an RRM; evidence that the dsRBD domain is not required for RNA binding but may be involved in substrate specificity; and LC/ESI-MS/MS data supporting a model in which DND1 function occurs in RNP granules. Although there exist several structures of individual RRMs there are very few structures of tandem RRMs. It remains unclear how RRMs work in tandem to recognize an RNA substrate. This study provides a unique example of tandem RRM–RNA recognition. The finding that the first RRM (RRM1) has an extended surface at the N-terminus has been observed previously in a related protein, and the authors provide strong evidence that this atypical RRM is conserved among other RNA-binding proteins. The DND1 RRM2 contributes to binding in a very unusual manner that has not been observed previously in RRMs. However, it is unclear whether this binding mode is present only in DND1 due to its lack of an RNP1 and RNP2 surface, if it is conserved in other proteins, or if it binds and functions this way in vivo. The lack of apparent conservation and importance of this particular binding mode on function dampens enthusiasm somewhat. The in cellulo data nicely supports the structural findings, but is limited and could be further strengthened to support the claims made. Overall, this manuscript presents a significant contribution to the RNA-protein structural biology field, informs on how RRMs work in tandem to bind RNA substrates, and provides insights into DND1 biology.

Major comments:

- The RRM2-RNA interactions may simply be due to non-specific/electrostatic interactions and may not be physiologically relevant. The authors do not present in cellulo evidence to support the importance of RRM2-RNA interactions to function. The other RNA-binding proteins included in the multiple sequence alignment contain RNP1 and RNP2 sequences, and the authors acknowledge that the other proteins may have a more conventional binding interface.
- It is clear that this protein is extremely challenging to work with, and the authors have overcome significant obstacles to determine the structure of the tandem RRM-RNA complex. The determination of this structure is a major achievement, and is a significant contribution to the field.
- The experimental replicates or details of statistical analysis are not included for the ITC data in the methods or main text. The authors should include this information, particularly for constructs with weak affinity.

- The IP and LC-MS data indicating DND1 is present in RNP granules is interesting, but the evidence remains weak. The claim would be strengthened and validated by fluorescence microscopy methods to visualize DND1 localization.

Minor comments:

- Figure S7C is mislabeled; DND1 does not have W207 or K189 residues. This data for the figure and main text should be reviewed and updated.

- The negative control in Figure 1 is also effected by DND1 mutations, despite not being a target of DND1. The authors should explain why the negative control is altered by DND1 mutations.

- Figure S6 could be made more clear by annotating with secondary structure features to more clearly see atypical eRRM features.

- The description of alpha helices on page 15 (section “structural extension of the eRRM1 increases RNA binding affinity and stabilizes a backbone turn in the recognized RNA) is confusing to the reader. It is stated that the DND1 N-terminal extension has a third alpha-helix, which is assumed to be alpha 0 in RRM1. Later in the section, it is stated that “our structure reveals that this beta-hairpin packed on a third alpha-helix is essential for increasing the affinity to the RNA by fixing the backbone ... (Figs 4D, E)” however, no evidence has been presented to demonstrate that the alpha 0 helix has any effect on affinity. Figure 4D and 4E show the alpha 2 helix of RRM2, not the alpha 0 helix of RRM1. It would be beneficial for the reader for the authors to clarify this section and present evidence to support the claims regarding affinity.

Reviewer #2 (Remarks to the Author):

Using a combination of structural biology and in cellulo approaches, the manuscript by Duszczuk et al. investigates principles of RNA target recognition and regulation of the germ-line specific RNA-binding protein (RBP) Dead End 1 (DND1). As DND1 has a critical function in germ cell development and its truncation is associated with formation of testicular teratomas in mice, it is key to determine the exact molecular mechanism of DND1-mediated target recognition and regulation. DND1 is likely to exert its functions by its unique domain structure that consist of two tandem single-stranded RNA recognition motifs (RRM1 and RRM2) followed by a double stranded RNA binding domain (dsRBD). The authors investigated functional contribution of the individual DND1 domains and proposed a model in which the two tandem RRMs are responsible for RNA target binding while dsRBD contributes to the target regulation. The authors show that DND1 exerts its RNA binding function by two RRMs building an unusual binding pocket.

In my opinion, the study advances our understanding of how the key regulator of germ line development DND1 achieves its target binding specificity and suggests the molecular mechanism for its unusual target recognition. My specific comments to clarify certain aspects of the manuscript are summarized below in a section-by-section form.

1. DND1 binds CRIP/RIP targets in cellulo mainly through its RRM1

The authors perform RNA-IP followed by RT-qPCR of two DND1 mutants to assess the contribution of individual DND1 domains to the binding of two mRNA targets, PMAIP1 and SMN, previously identified by PAR-CLIP and DO-RIP (Kedde et al., 2007; Yamaji et al., 2017). Based on the results shown in Figure 1B and S3A, the authors concluded that DND1 interacts with its targets in cellulo mainly through RRM1, while the dsRBD is dispensable for the interactions. The authors suggest that the contribution of RRM2 “could not be tested easily, as its RNA interaction surface could not be predicted considering the non-canonical nature of this RRM.”

Minor points:

- In Figure 1B and S3A, the authors should provide statistical information on significance of their RNA-IP experiments.
- In Figure S3A, the overall binding of all mutants seems to be several folds lower in comparison to RIP2 and RIP1 (Fig 1). This experiment should be repeated to further support the results of RIP1 and RIP2 or the authors should explain the difference in individual experiments.
- In Figure S3B, the authors should add a Western blot of RNA-IP whole cell extracts with an appropriate loading control (i.e., actin or GAPDH).
- When giving rationale for choosing PMAIP1 and SMN mRNAs as targets for RNA-IP analysis, the authors could clarify the number and distribution of putative motifs identified by PAR-CLIP and DO-RIP, respectively. In addition, the authors could have added p27 mRNA in these experiments for consistency throughout the paper (p27 fragment is used in the next part of the results section).

2. DND1's tandem RRMs cooperatively bind AU-rich RNA with high affinity

The authors perform series of ITC and NMR experiments to investigate the individual contributions of RRM1 and RRM2 in RNA binding of short oligonucleotides derived from the p27 transcript, a well-established DND1 target (Kedde et al., 2007). Based on these data, the authors concluded that the RNA binding is executed primarily by RRM1, while RRM2, which does not bind any oligonucleotide sequences alone, has a supportive role.

Major points:

- In Figure 2, the authors show ITC measurements of either only RRM1 or the DND1 1-235 mutant containing two RRMs to CUUAAUUUG. Based on their results, the authors concluded that the presence of both RRMs is required for efficient RNA binding. Have the authors excluded the possibility that the lack of RRM1 binding alone to the same oligonucleotide could be a consequence of protein insolubility? Have the authors considered performing ITC measurements with DND1 1-235 mutant carrying point mutations in RRM1 and/or RRM2 to dissect contribution of each RRM? In addition, how have the authors bypassed poor solubility of the RRM12 protein, which was the main reason for protein purification for the solution structure was completed in complex with the AU-rich oligonucleotide (page 7)?

- DO-RIP analysis of DND1 binding, published by Yamaji and colleagues, identified the binding to NYAYUNN motif. In this part of the results section, the authors only tested three AU-rich sequences which contain small variability. The p27 mRNA has 47 NYAYUNN motifs with varying degrees of C and G nucleotides. Have the authors also tested the binding to more CG-rich sequences? If not, the statement of testing the NYAYUNN motif should be removed from the summary at the end of the introduction.

3. Mutation of key RNA binding residues compromises the RNA interaction in vitro

- The authors identified K197A and W215F mutants in the RRM2 that result in weaker binding to RNA in vitro, but do not further investigate the effect of these point mutations individually or combined in vivo by RNA-IP (similar to the experiments in the Results section 1). Is there any specific reason why these experiments were not done?

- Moreover, the Figure S7C (related to Figure 4) shows the same WT ITC measurements from Figure 2, as well as ITC measurements of W207F and K189A mutants, while the text on page 10 related to these data does not mention these mutants. Instead, the authors describe results for K197A and W215F mutants, whose ITC data (referred to in the text to be in Figure S7C) are not shown. The authors should clarify these points.

4. Introduction of an AU-rich motif into a reporter gene 3'UTR is necessary and sufficient for target repression by DND1

To investigate the function of RNA binding to DND1 activity, the authors employ a previously published luciferase-based assay (Kedde et al., 2007). Briefly, the assay consists of the expression of a luciferase-based reporter carrying a partial p27 3'UTR downstream of the Renilla luciferase ORF in HEK293T cells. Due to the mild luciferase repression by miR-221 in the presence of FLAG-tagged DND1, the authors change the strategy and insert a UAUUU pentamer alone or two UAUUU pentamers spaced by a UUUU tetramer into the TERT 3' UTR to directly test the contribution of RNA binding to the designed motif.

The insertion of two pentamers increased the luciferase activity in the presence of DND1 compared to the control assay without pentamers, or the presence of 1-235 DND1 mutant, suggesting that the RRM1-2 are not sufficient for target regulation.

Minor point:

- The experiment with two AU motifs is convincing. However, it is not clear why the authors chose the insertion of pentamers spaced by the U tetramer, instead of inserting two copies of a motif identified in previous experiments.

5. The interactome of DND1 targets in HEK293T is enriched for mRNA stabilizing proteins and proteins present in RNP granules

To understand the biological effect of DND1 binding, the authors transfected either wild type DND1, R98A mutant, or the 1-235 truncation mutant, and performed an IP and mass spectrometry analysis. The authors identified 20 and 25 proteins at statistically different levels in R98A and 1-235 truncation compared to wild type DND1, 29 of which were identified in a previous study (Yamaji et al., 2017). Following the GO analysis, the authors focused on the mRNA stabilizing proteins, from which they validated the interaction with ELAVL1 and concluded that the DND1-dependent target regulation occurs in an RNP granule-like environment.

Specific points:

- As the authors do not provide any additional functional data to support the MS results beyond the direct interaction with ELAVL1 (i.e., no fluorescence microscopy of DND1 and ELAVL1; the same is valid for TRAP1 interaction and mitochondrial targeting), the section has a speculative and discussion-like character and should be shortened and/or moved to the Discussion.

Other minor points:

- Ding et al., 1999 cited in Figure S1. not listed in references
- For easier presentation, the sequences and corresponding color scheme could be indicated, rather than color-coded in Figure S4.
- Figures S1, S2, and S6 contain related data and could therefore be fused in one figure

Reviewer #3 (Remarks to the Author):

In the present article entitled “The solution structure of Dead End bound to AU-rich RNA reveals an unprecedented mode of tandem RRM-RNA recognition required for mRNA regulation” Duszczuk et al. depict how DND1 recognizes and represses its cellular targets at the molecular level using different approaches including NMR structure analysis and immunoprecipitation coupled with liquid chromatography mass spectrometry. The investigations are robust, the manuscript is well-written and figures illustrates perfectly the obtained results. Nonetheless, before publication in Nature Communications, I suggest doing some minor edits. Here, I detail my suggestions (and questions) point-by-point:

- Introduction: The introduction is well written but I feel that the last section where what is done is explained is too extensive for an introduction. I would suggest to briefly mention what it is done but without intricate details about the obtained results.
- Results: in the proteomics study, were those 769 proteins detected in all cases in three technical replicates consistently? Or that number includes all detected proteins considering all the samples and all the replicates without filtering by consistency of identification?
- 25 and 20 proteins were differentially present in R98A mutant and the 1-235 dsRBD truncation mutant pulldowns, but for that analysis authors used t-test with FDR = 10%. I feel that this percentage of FDR is too high. Why authors have used that FDR correction?
- Why authors have used a 1.5 FC threshold? Is there any statistical meaning behind this number?
- Figure 5. I don't think it is necessary to include p-values that are higher than 0.05 in the graphics. I would suggest to just include p-values for comparisons with statistical significance.
- In figure 6. Maybe it is due to the resolution of the figure in the compiled pdf file, but I don't understand that “** The column left of the y-axis shows proteins present in WT but completely missing in the mutants.” Maybe the list of this missing proteins can be detailed in a supplementary table? Or in another more illustrative way?
- Method details. There is a dot in the title RNA Samples. Remove it to be consistent with other titles.
- Transfections and Dual luciferase activity analysis. Remove the capital letter from Dual.
- Write the numbers in subscript form NH_4CO_3 . Revise all the document as I detected other chemical compounds with the same problem.
- Use consistent names for reagents: ej. ACN and acetonitrile in “150 μ l of 0.1%TFA/50% acetonitrile” and “150 μ l of 0.1%TFA/50% acetonitrile”. Revise the entire document.
- Authors did not mention any parameter or detail about the LC-MS/MS analysis. Please include sufficient information about LC and MS instruments, LC gradient, MS parameters,...
- I don't understand the following sentence: Only proteins with a median of exclusive unique spectral counts larger than two in the DND1 WT pulldown were compared.

The solution structure of Dead End bound to AU-rich RNA reveals an **unprecedented mode of tandem RRM-RNA recognition required for mRNA regulation**

Response to Referees

Reviewer #1 (Remarks to the Author):

Summary:

In this manuscript, the authors determine the solution structure of the tandem RRM domains of the dead end (DND1) protein bound to RNA. In addition, in cellulo evidence supports the structural findings. Key findings include an unprecedented mode of RNA recognition by an RRM; evidence that the dsRBD domain is not required for RNA binding but may be involved in substrate specificity; and LC/ESI-MS/MS data supporting a model in which DND1 function occurs in RNP granules. Although there exist several structures of individual RRMs there are very few structures of tandem RRMs. It remains unclear how RRMs work in tandem to recognize an RNA substrate. This study provides a unique example of tandem RRM–RNA recognition. The finding that the first RRM (RRM1) has an extended surface at the N-terminus has been observed previously in a related protein, and the authors provide strong evidence that this atypical RRM is conserved among other RNA-binding proteins. The DND1 RRM2 contributes to binding in a very unusual manner that has not been observed previously in RRMs. However, it is unclear whether this binding mode is present only in DND1 due to its lack of an RNP1 and RNP2 surface, if it is conserved in other proteins, or if it binds and functions this way in vivo. The lack of apparent conservation and importance of this particular binding mode on function dampens enthusiasm somewhat. The in cellulo data nicely supports the structural findings, but is limited and could be further strengthened to support the claims made. Overall, this manuscript presents a significant contribution to the RNA-protein structural biology field, informs on how RRMs work in tandem to bind RNA substrates, and provides insights into DND1 biology.

We thank the reviewer for the recognition of the unique aspects of our structure and the contribution of our work to the protein-RNA structural field. The unique binding mode identified in this study is not conserved between the different proteins throughout the entire hnRNPR-like family. Nevertheless, the almost full conservation between species supports its utmost importance and reinforces the need to study this binding mode for the unique function of the DND1 protein.

Major comments:

- The RRM2-RNA interactions may simply be due to non-specific/electrostatic interactions and may not be physiologically relevant. The authors do not present in cellulo evidence to support the importance of RRM2-RNA interactions to function. The other RNA-binding proteins included in the multiple sequence alignment contain RNP1 and RNP2 sequences, and the authors acknowledge that the other proteins may have a more conventional binding interface.

Our structure shows that both electrostatic interactions and H-bonding contributes to the RRM2-RNA binding pocket. For example, the U6 base carbonyl O4 is hydrogen-bonded by both the NH₃ of K196 and the conserved sidechain HE1 of W215 (Fig. 4D). We presented evidence of weakened RNA binding of the W215F mutant in ITC and NMR titrations (Fig. S7). We tried to test several RRM2 binding pocket mutants in our RIP assays, unfortunately, obtaining irreproducible results, presumably because these mutants, which we know to destabilize the protein *in vitro*, also render the overexpressed protein unstable in cells and therefore there is high variability between the replicates depending on sample handling. We would like to stress that the physiological relevance of this pocket for DND1 function has been confirmed in functional studies in zebrafish where the equivalent to the K197 mutant (K200T) was the only shotgun mutant outside of RRM1 causing loss of function (Slanchev et al. 2009). The fact that the other proteins in the hnRNPR-like family have canonical RNP sequences in RRM2 does not exclude that next to canonical binding, the novel binding mode of DND1 RRM2 may be used in addition. However, how the multiple RRMs of the other family members cooperate in binding RNA remains to be discovered and is beyond the scope of this paper.

- It is clear that this protein is extremely challenging to work with, and the authors have overcome significant obstacles to determine the structure of the tandem RRM-RNA complex. The determination of this structure is a major achievement, and is a significant contribution to the field.

We thank the reviewer for recognizing the challenges in determining the DND1 RRM12-RNA structure. Indeed, difficulties in working with DND1 *in vitro* led to resorting to modeling approaches to address DND1's RNA binding properties, the limitations of which resulted in incorrect structural models (Slanchev et al. 2009). We would like to highlight that the challenges working with WT DND1 were further compounded in mutants of the protein. Despite this we were able to produce a number of mutants to use in validation of the protein-RNA interactions in biochemical and cell-based approaches.

- The experimental replicates or details of statistical analysis are not included for the ITC data in the methods or main text. The authors should include this information, particularly for constructs with weak affinity.

Although we did perform some of the ITC's more than once, the ITC experiments presented in this paper are single runs and the errors in the parameters reflect the data fitting process, fitting to a one-site binding model. More information on the fitting procedure has now been added to the methods section. Especially for the purpose in this paper, which is mainly comparing binding between single and double RRMs, and U-rich and AU-rich RNAs (where we show 30- to 80-fold affinity differences), as well as WT *versus* mutant protein (4- to 10-fold affinity differences), we deem this sufficient. ITC has limitations in measuring precise binding affinities. Thus, for the weaker complexes, we stated the Kd as > being larger to (a fit value).

In addition, we corrected an error in the methods and SI concerning the ITC buffer composition: since the earliest NMR titrations and ITC experiments, we changed the buffer from MES+NaCl to phosphate. The ITC runs that are included in this paper were performed in phosphate and this has now been correctly stated everywhere in the paper.

- The IP and LC-MS data indicating DND1 is present in RNP granules is interesting, but the evidence remains weak. The claim would be strengthened and validated by fluorescence microscopy methods to visualize DND1 localization.

We have now performed novel fluorescence microscopy experiments. We observed a diffused staining of the protein in both nuclear and cytoplasmic compartments, arguing against them getting enriched in RNP granules. We added these data to the supplemental information (Fig. S8). Treating the cells with 1,6 hexanediol, used to dissolve liquid-liquid phase separation (LLPS) does not change DND1 localization (added below here as Fig. S8D for your information only), showing that at least in HEK293T DND1 is not localized in RNP granules. In addition, we have now weakened our claims in the revised manuscript, moving the hypothesis that based on the interacting proteins found in our IPs by MS, DND1 could be localized in RNP granules in the germline stem cells, to the discussion. As the HEK293T IPs and microscopy data complement the DND1 interaction data presented by Yamaji et al. (Yamaji et al. 2017), we are convinced that these data represent a valuable addition to the DND1 literature.

S8D

Figure S8D: Indirect immunofluorescence (IF) experiments in HEK293T cells transiently transfected with plasmids encoding the indicated FLAG tagged Dnd1 proteins treated 5 minutes with 5% 1,6 Hexanediol or left untreated (notrt). FLAG in green and DAPI stained nuclei are in blue. Scale bar 10 μ m.

Minor comments:

- Figure S7C is mislabeled; DND1 does not have W207 or K189 residues. This data for the figure and main text should be reviewed and updated.

We thank the reviewer for noticing this mistake, we have relabeled this figure properly.

- The negative control in Figure 1 is also affected by DND1 mutations, despite not being a target of DND1. The authors should explain why the negative control is altered by DND1 mutations.

The negative control RNA has some residual unspecific binding. Although the R98A mutant fails to bind short RNA *in vitro* as shown by NMR titrations, it also shows some residual enrichment in this experiment. Our new analysis of all IP data (Fig. 1B) shows that the difference of the enrichment between the targets pulled down by the R98A mutants and the Dnd1WT pulldown of the negative control is not significant (ns).

- Figure S6 could be made more clear by annotating with secondary structure features to more clearly see atypical eRRM features.

We thank the reviewer for this suggestion and now added the secondary structure features to Figure S6.

- The description of alpha helices on page 15 (section “structural extension of the eRRM1 increases RNA binding affinity and stabilizes a backbone turn in the recognized RNA) is confusing to the reader. It is stated that the DND1 N-terminal extension has a third alpha-helix, which is assumed to be alpha 0 in RRM1. Later in the section, it is stated that “our structure reveals that this beta-hairpin packed on a third alpha-helix is essential for increasing the affinity to the RNA by fixing the backbone ... (Figs 4D, E)” however, no evidence has been presented to demonstrate that the alpha 0 helix has any effect on affinity. Figure 4D and 4E show the alpha 2 helix of RRM2, not the alpha 0 helix of RRM1. It would be beneficial for the reader for the authors to clarify this section and present evidence to support the claims regarding affinity.

We apologize for the confusion. The N-terminal helix alpha0 forms part of the core of the eRRM, interacting with alpha2 and the N-terminal beta hairpin. It is the residues on the beta-hairpin (N37; Q39; R88 as described in the text) that increase affinity by charge interactions with the RNA backbone. Obtaining biophysical experimental evidence of the contribution of the N-terminal extension was not possible: an N-terminal truncation mutant of either alpha 0 or alpha0-beta-1/0, as well as point mutants of the N-terminal beta-hairpin render the protein insoluble.

We now changed the text from: ‘ β -hairpin packed on a third α -helix’ to ‘N-terminal extension’ for clarity.

Reviewer #2 (Remarks to the Author):

Using a combination of structural biology and in cellulo approaches, the manuscript by Duszczuk et al. investigates principles of RNA target recognition and regulation of the germ-line specific RNA-binding protein (RBP) Dead End 1 (DND1). As DND1 has a critical function in germ cell development and its truncation is associated with formation of testicular teratomas in mice, it is key to determine the exact molecular mechanism of DND1-mediated target recognition and regulation. DND1 is likely to exert its functions by its unique domain structure that consist of two tandem single-stranded RNA recognition motifs (RRM1 and RRM2) followed by a double stranded RNA binding domain (dsRBD). The authors investigated functional contribution of the individual DND1 domains and proposed a model in which the two tandem RRMs are responsible for RNA target binding while dsRBD contributes to the target regulation. The authors show that DND1 exerts its RNA binding function by two RRMs building an unusual binding pocket.

In my opinion, the study advances our understanding of how the key regulator of germ line development DND1 achieves its target binding specificity and suggests the molecular mechanism for its unusual target recognition. My specific comments to clarify certain aspects of the manuscript are summarized below in a section-by-section form.

1. DND1 binds CRIP/RIP targets in cellulo mainly through its RRM1

The authors perform RNA-IP followed by RT-qPCR of two DND1 mutants to assess the contribution of individual DND1 domains to the binding of two mRNA targets, PMAIP1 and SMN, previously identified by PAR-CLIP and DO-RIP (Kedde et al., 2007; Yamaji et al., 2017). Based on the results shown in Figure 1B and S3A, the authors concluded that DND1 interacts with its targets in cellulo mainly through RRM1, while the dsRBD is dispensable for the interactions. The authors suggest that the contribution of RRM2 “could not be tested easily, as its RNA interaction surface could not be predicted considering the non-canonical nature of this RRM.”

Minor points:

- In Figure 1B and S3A, the authors should provide statistical information on significance of their RNA-IP experiments.

Please note that each of the figures represents an independent biological RIP experiment. It is well known that IP efficiencies may vary between independent experiments, even using the same antibody, introduced by small sample handling differences during the protocol that are difficult to control for. Before calculating statistical significance on such set of experiments, these efficiencies should be in principle normalized for. This is what we now did and present in the new Fig. 1B. In total we ran five independent IP experiments with technical triplicates of each sample. As we deem only data with SD (deltaCt) < 0.3 reliable for the averaged data point, we could only use partial data from each experiment. We chose to present three of these experiments independently in the first version of this manuscript. We did not calculate statistical significance based on the different deltaCt values between the mutants for each plot/RIP individually, as this concerned RT-qPCR technical triplicates. The error bars in these figures represent the SD (deltaCt) of the technical triplicates of which the average is treated as one data point.

We now re-analyzed ALL IPs and normalized each to the enrichment of SMN1 pulled down by the WT-DND1 (see new Fig. 1B). We averaged three technical triplicates only if SD (deltaCt) < 0.3 and only present data of which at least N=3 data points / biological replicates of the IP was available. This analysis revealed one more target of DND1 (SRSF2) with sufficient data points that we now include in the manuscript. Interestingly, the dsRBD DOES contribute to binding of SRSF2. We rewrote the relevant sections accordingly: apparently the dsRBD only contributes to the binding of a subset of DND1 targets.

- In Figure S3A, the overall binding of all mutants seems to be several folds lower in comparison to RIP2 and RIP1 (Fig 1). This experiment should be repeated to further support the results of RIP1 and RIP2 or the authors should explain the difference in individual experiments.

As explained above, irrespective of the lower RIP efficiency, RIP 3 overall confirms the results of RIP1 and RIP2. The differences in efficiency are expected. The two added datasets in Fig. 1B confirm the results from RIP 1-3.

- In Figure S3B, the authors should add a Western blot of RNA-IP whole cell extracts with an appropriate loading control (i.e., actin or GAPDH).

Figure S3B is the whole-cell extract subjected to IP. The western blot and Coomassie staining concern the identical gel. The Coomassie staining serves as the loading control: Equal bands for all cell culture proteins across mutants show that equal volumes of whole cell extracts for the different mutants were loaded.

- When giving rationale for choosing PMAIP1 and SMN mRNAs as targets for RNA-IP analysis, the

authors could clarify the number and distribution of putative motifs identified by PAR-CLIP and DO-RIP, respectively. In addition, the authors could have added p27 mRNA in these experiments for consistency throughout the paper (p27 fragment is used in the next part of the results section).

P27 was chosen as a DND1 target in the luciferase assay to replicate the assays in the paper by Kedde et al. Indeed, we also intended to use p27 as a target in the RNA-IP analysis initially, but we were unsuccessful to detect it by RT-qPCR after pull down of DND1. One explanation could be that the p27/CDKN1B level in HEK293T is relatively low (rel. level 18 as measured by RNAseq by Yamaji et al.), moreover with a short half-life even though it contains 10 UAUU motifs in its 3'UTR. It should be noted that Ruthig et al. reported p27 as a target, reproducible in both DO-RIPs, including DATBAW motifs in the 3'UTR.

We chose PMAIP1 with an order of magnitude (level 115) higher mRNA levels because it had the top NXPM in the 3'UTR in the CLIP by Yamaji et al. It contains 15 UAUU motifs in its 3'UTR. Ruthig et al. reported PMAIP1 in both DO-RIPs for PMAIP1, including DATBAW motifs in the 3'UTR.

SRSF2 has the third-highest NXPM in the 3'UTR in Yamaji et al. and is expressed at the same level as PMAIP. It has 8-11 UAUU motifs in its 3'UTR, depending on the transcript. Ruthig et al. reported SRSF2 as a target, reproducible in both DO-RIPs, including DATBAW motifs in the 3'UTR.

SMN has mRNA levels in HEK293 in the same order of magnitude as p27 (SMN1: 25.3; SMN2 13.8) with 4-6 UAUU motifs in the 3'UTR, depending on the transcript variant. Ruthig et al. who reported SMN1/2 mRNAs as targets in DO-RIP, including DATBAW motifs in the 3'UTR. Interestingly, SMN had no reads in the Yamaji et al. paper. Please note that we changed the naming of SMN1 to SMN in the whole manuscript as we realized our primers are specific to both SMN1 and SMN2 mRNAs. Finally, we chose SLC25A6 as a negative control because of comparable RNAseq levels to PMAIP1 but no reads in either Yamaji et al. or Ruthig et al, even though it contains one UAUU motif in its 3'UTR.

2. DND1's tandem RRM cooperatively bind AU-rich RNA with high affinity

The authors perform series of ITC and NMR experiments to investigate the individual contributions of RRM1 and RRM2 in RNA binding of short oligonucleotides derived from the p27 transcript, a well-established DND1 target (Kedde et al., 2007). Based on these data, the authors concluded that the RNA binding is executed primarily by RRM1, while RRM2, which does not bind any oligonucleotide sequences alone, has a supportive role.

Major points:

- In Figure 2, the authors show ITC measurements of either only RRM1 or the DND1 1-235 mutant containing two RRMs to CUUAUUUG. Based on their results, the authors concluded that the presence of both RRMs is required for efficient RNA binding. Have the authors excluded the possibility that the lack of

RRM1 binding alone to the same oligonucleotide could be a consequence of protein insolubility? Have the authors considered performing ITC measurements with DND1 1-235 mutant carrying point mutations in RRM1 and/or RRM2 to dissect contribution of each RRM? In addition, how have the authors bypassed poor solubility of the RRM12 protein, which was the main reason for protein purification for the solution structure was completed in complex with the AU-rich oligonucleotide (page 7)?

The RRM1 and RRM12 constructs precipitate rapidly under high concentrations necessary for structure determination by NMR spectroscopy (above 0.5mM, low salt) which is not compatible with the up to three days lasting experiments. At concentrations necessary for NMR titrations (0.1 mM) and ITC (in the order of 0.01 mM) it is stable for some time if it is freshly prepared. We can see in the NMR spectra that after the binding experiment the majority of the soluble protein present at the start of the experiment is still soluble because the peak intensity is not reduced (Fig. S4B).

In Figure S7B we do exactly the experiment proposed by the reviewer: we test binding of point mutants using NMR and in S7C using ITC to dissect contribution of each RRM. Single mutants of RRM1 abolish binding (M90A, R98A) and single mutants of RRM2 weaken it (K197A; W215F).

To produce an NMR sample under conditions suitable to solve the structure (low salt, high concentration) we bypassed poor solubility by always preparing the RRM12 protein fresh, under high salt conditions and then exchanging into low salt conditions quickly in the presence of RNA before a final concentration step.

- DO-RIP analysis of DND1 binding, published by Yamaji and colleagues, identified the binding to NYAYUNN motif. In this part of the results section, the authors only tested three AU-rich sequences which contain small variability. The p27 mRNA has 47 NYAYUNN motifs with varying degrees of C and G nucleotides. Have the authors also tested the binding to more CG-rich sequences? If not, the statement of testing the NYAYUNN motif should be removed from the summary at the end of the introduction.

We would like to clarify that Yamaji found enrichment of a UUU/UUA triplet using PAR-CLIP of DND1 and Ruthig et al. using the DO-RIP-seq approach described [A/G/U]AU[C/G/U]A[A/U] enriched in DND1 targets. The statement 'DO-RIP analysis of DND1 binding, published by Yamaji and colleagues, identified the binding to NYAYUNN motif' written here is not correct. Our structure, published in this manuscript has allowed us to deduce recognition of an NYAYUNN motif based on the specific interactions we see between protein and RNA as described in the 'Structural details: specific readout by DND1's tandem RRMs' section. Our starting point were the U-rich and AU-rich regions defined by Kedde et al. as DND1 interaction sites in p27. We were able to define a central position of an adenosine necessary for high affinity binding in vitro with a

biophysical approach. We have not performed extensive binding assays using RNA with varying degrees of G and C nucleotides in the N and Y positions. It remains to be discovered if G and C nucleotides in the N and Y positions would change the affinity and possibly the hydrogen bonding network to DND1 RRM12. The structural work necessary to answer such question is beyond the scope of the work presented in this manuscript.

3. Mutation of key RNA binding residues compromises the RNA interaction *in vitro*

- The authors identified K197A and W215F mutants in the RRM2 that result in weaker binding to RNA *in vitro*, but do not further investigate the effect of these point mutations individually or combined *in vivo* by RNA-IP (similar to the experiments in the Results section 1). Is there any specific reason why these experiments were not done?

We know from *in vitro* experiments that the single RRM2 K197A and W215F mutants weaken RNA binding while the R98A RRM1 mutant abolishes it completely. Therefore, we also did expect a smaller IP effect on RRM2 mutants. We have done the IP using the K197A mutant but got irreproducible results over the different IP's. This is likely because of the instability of the mutants. As the stability *in vitro* of the W215F mutant was much lower than the K197A mutant, we did not test W215F or a double RRM2 binding pocket mutant in IP.

- Moreover, the Figure S7C (related to Figure 4) shows the same WT ITC measurements from Figure 2, as well as ITC measurements of W207F and K189A mutants, while the text on page 10 related to these data does not mention these mutants. Instead, the authors describe results for K197A and W215F mutants, whose ITC data (referred to in the text to be in Figure S7C) are not shown. The authors should clarify these points.

We would like to apologize for this labeling mistake that has now been corrected to W215F and K197A. During the writing of the manuscript we changed residue numbering from numbering in the RRM12 construct to the full-length numbering which amounted to (+8).

4. Introduction of an AU-rich motif into a reporter gene 3'UTR is necessary and sufficient for target repression by DND1

To investigate the function of RNA binding to DND1 activity, the authors employ a previously published luciferase-based assay (Kedde et al., 2007). Briefly, the assay consists of the expression of a luciferase-based reporter carrying a partial p27 3'UTR downstream of the Renilla luciferase ORF in HEK293T cells. Due to the mild luciferase repression by miR-221 in the presence of FLAG-tagged DND1, the authors

change the strategy and insert a UAUUU pentamer alone or two UAUUU pentamers spaced by a UUUU tetramer into the TERT 3' UTR to directly test the contribution of RNA binding to the designed motif. The insertion of two pentamers increased the luciferase activity in the presence of DND1 compared to the control assay without pentamers, or the presence of 1-235 DND1 mutant, suggesting that the RRM1-2 are not sufficient for target regulation.

Minor point:

- The experiment with two AU motifs is convincing. However, it is not clear why the authors chose the insertion of pentamers spaced by the U tetramer, instead of inserting two copies of a motif identified in previous experiments.

The rationale was to create an RNA where potentially two copies of the protein could bind. To prevent steric hindrance a short spacer between the motifs was introduced. With a short linker, the avidity effect is still possible if only one protein would bind. We added this rationale to the text.

5. The interactome of DND1 targets in HEK293T is enriched for mRNA stabilizing proteins and proteins present in RNP granules

To understand the biological effect of DND1 binding, the authors transfected either wild type DND1, R98A mutant, or the 1-235 truncation mutant, and performed an IP and mass spectrometry analysis. The authors identified 20 and 25 proteins at statistically different levels in R98A and 1-235 truncation compared to wild type DND1, 29 of which were identified in a previous study (Yamaji et al., 2017). Following the GO analysis, the authors focused on the mRNA stabilizing proteins, from which they validated the interaction with ELAVL1 and concluded that the DND1-dependent target regulation occurs in an RNP granule-like environment.

Specific points:

- As the authors do not provide any additional functional data to support the MS results beyond the direct interaction with ELAVL1 (i.e., no fluorescence microscopy of DND1 and ELAVL1; the same is valid for TRAP1 interaction and mitochondrial targeting), the section has a speculative and discussion-like character and should be shortened and/or moved to the Discussion.

We have now added fluorescence microscopy data on ELAVL1 (Supp. Fig. 8B) and moved the speculative part to the discussion.

Other minor points:

- Ding et al., 1999 cited in Figure S1. not listed in references

- For easier presentation, the sequences and corresponding color scheme could be indicated, rather than color-coded in Figure S4.
- Figures S1, S2, and S6 contain related data and could therefore be fused in one figure

We have added the reference, followed the suggestion on presentation of figure S4 and combined figures S1/2/6 into one figure (and updated the text accordingly).

Reviewer #3 (Remarks to the Author):

In the present article entitled “The solution structure of Dead End bound to AU-rich RNA reveals an unprecedented mode of tandem RRM-RNA recognition required for mRNA regulation” Duszczuk et al. depict how DND1 recognizes and represses its cellular targets at the molecular level using different approaches including NMR structure analysis and immunoprecipitation coupled with liquid chromatography mass spectrometry. The investigations are robust, the manuscript is well-written and figures illustrates perfectly the obtained results. Nonetheless, before publication in Nature Communications, I suggest doing some minor edits. Here, I detail my suggestions (and questions) point-by-point:

- Introduction: The introduction is well written but I feel that the last section where what is done is explained is too extensive for an introduction. I would suggest to briefly mention what it is done but without intricated details about the obtained results.

We have now shortened the last section of the introduction.

- Results: in the proteomics study, were those 769 proteins detected in all cases in three technical replicates consistently? Or that number includes all detected proteins considering all the samples and all the replicates without filtering by consistency of identification?

The 769 proteins include all detected proteins considering all samples and replicates. Table S4 clearly shows which proteins were discovered in which of the replicates and mutants. We have added the words ‘over all mutants and replicates’ to this number for clarity.

- 25 and 20 proteins were differentially present in R98A mutant and the 1-235 dsRBD truncation mutant pulldowns, but for that analysis authors used t-test with FDR = 10%. I feel that this percentage of FDR is too high. Why authors have used that FDR correction?

With a 10% FDR we obtained 25 discoveries in the R98A mutant analysis and 20 discoveries in the 1-235 mutant analysis. A 5% FDR gave the top 18 and 10 discoveries of this list, respectively. Even though raising the FDR to 10% could add one or two false discoveries to the top list, we feel this is acceptable as it increases the informative value of the found protein interaction network, especially for the 1-235 dsRBD truncation mutant where the discoveries are doubled.

Please note that we corrected a typo in Table S4: the values in the T-test sheet were obtained with the 10%, not 5%FDR.

- Why authors have used a 1.5 FC threshold? Is there any statistical meaning behind this number?

1.5 FC is considered a moderately strict cutoff for differential interactions. Again, we feel that especially in the case of the interaction network of the 1-235 dsRBD truncation mutant, using a moderately strict cutoff as opposed to a stricter cutoff of 2FC, increases the informative value of the experiment. Of course, the reader may judge the results based upon their preferred FC cutoff.

- Figure 5. I don't think it is necessary to include p-values that are higher than 0.05 in the graphics. I would suggest to just include p-values for comparisons with statistical significance.

In our opinion including all p-values is informative to illustrate the difference between the mut1 single AU-rich binding site (lack of statistically significant effect of mutations) and the mut2 double AU-rich binding site (statistically significant effect of mutations) that is not clear just by looking at the figure without explicit p-values.

- In figure 6. Maybe it is due to the resolution of the figure in the compiled pdf file, but I don't understand that "*** The column left of the y-axis shows proteins present in WT but completely missing in the mutants." Maybe the list of this missing proteins can be detailed in a supplementary table? Or in another more illustrative way?

We would prefer to keep the figure as is as it illustrates the p-value of the enrichment difference that is stated in Table S4.

- Method details. There is a dot in the title RNA Samples. Remove it to be consistent with other titles.

We have corrected this typo.

- Transfections and Dual luciferase activity analysis. Remove the capital letter from Dual.

We have corrected this typo.

- Write the numbers in subscript form NH_4CO_3 . Revise all the document as I detected other chemical compounds with the same problem.

We revised the document for correct chemical compound naming.

- Use consistent names for reagents: ej. ACN and acetonitrile in “150µl of 0.1%TFA/50% acetonitrile” and “150µl of 0.1%TFA/50% acetonitrile”. Revise the entire document.

We revised the document for reagent name consistency.

- Authors did not mention any parameter or detail about the LC-MS/MS analysis. Please include sufficient information about LC and MS instruments, LC gradient, MS parameters,...

The materials & methods section for the LC-MS/MS analysis has now been extended.

- I don't understand the following sentence: Only proteins with a median of exclusive unique spectral counts larger than two in the DND1 WT pulldown were compared.

We changed this sentence to ‘Proteins with less than two exclusive unique spectral counts (median of the triplicate IP) in the DND1 WT IP were excluded from the analysis’ for clarity.

References:

Slanchev, K. *et al.* Control of Dead end localization and activity--implications for the function of the protein in antagonizing miRNA function. *Mech. Dev.* **126**, 270–7 (2009).

Yamaji, M. *et al.* DND1 maintains germline stem cells via recruitment of the CCR4–NOT complex to target mRNAs. *Nature* **543**, 568–572 (2017).

REVIEWER COMMENTS

Reviewer #1 (Remarks to the Author):

The authors adequately responded to this reviewer's previous comments with the exception of the ITC data. This reviewer remains concerned that there are no replicates of the ITC experiments, particularly in light of the high variability between replicates in other experiments, attributed to sample handling. Replicates are essential to ensure that the findings are reproducible. In particular for wild type versus mutant proteins with 4-10 fold affinity differences, variance between replicates can reach 4-fold differences in some cases. The errors from the fit are likely an underestimate of the true error of the binding affinity, which is established through performing multiple replicates. The lack of demonstrated reproducibility for the ITC data requires further revision of the manuscript to perform technical replicates and provide a more robust estimate of the binding affinity and associated error.

Reviewer #2 (Remarks to the Author):

The authors fully addressed my comments and I believe the manuscript is now suitable for publication.

Reviewer #3 (Remarks to the Author):

The authors have successfully addressed all my previous concerns and I find the manuscript acceptable for publication.

Reviewer #1 (Remarks to the Author):

The authors adequately responded to this reviewer's previous comments with the exception of the ITC data. This reviewer remains concerned that there are no replicates of the ITC experiments, particularly in light of the high variability between replicates in other experiments, attributed to sample handling. Replicates are essential to ensure that the findings are reproducible. In particular for wild type versus mutant proteins with 4-10 fold affinity differences, variance between replicates can reach 4-fold differences in some cases. The errors from the fit are likely an underestimate of the true error of the binding affinity, which is established through performing multiple replicates. The lack of demonstrated reproducibility for the ITC data requires further revision of the manuscript to perform technical replicates and provide a more robust estimate of the binding affinity and associated error.

ITC data are presented only in two figures: Figure 2 and supplementary figure S5.

We will address the reviewer's and editor's concerns in 3 parts:

- 1) First, we have addressed the issue with the fluctuations in the binding curve at higher molar ratios in Figs. 2A and S5C (which are the same figure), for which a few points were removed from the curve before fitting in the original manuscript. These types of fluctuations are caused by air bubbles introduced during later injections and it is common to remove these data points as the integrals are complete outliers on the curve to be fit.

To prevent doubts about this part of the data, we have now removed all last 10 injections / data points. In fact, it would have been fine to stop the experiment earlier as the binding reaction has come to an end by then and the bottom, middle and top of the binding curve are visible and allow for an accurate fit.

Fitting these truncated data (just the first 30 injections) basically gives identical binding parameters that we have now adjusted in the paper. The last 10 points which include the air bubble-caused fluctuations are totally irrelevant for the fit in this case:

New fit (30 injections):

Old fit (40 injections, 2 points removed):

- 2) In Figs. 2C-E the binding curves show a low affinity. Repeating the ITC from panel C, D and E will not provide any additional information. The affinity of the single RRM1 to AU-rich and RRM12 to U-rich RNA is already very weak and the affinity can only be estimated with a lower estimate. Doing a replicate will not bring any additional quantitative information since no standard deviation could be obtained. **The main information in these figures is that the WT affinity to AU-rich RNA is in the range of 1 μM and that there is a 30-100 fold loss of affinity if not both RRMs or AU-rich RNA is present.**

In an earlier stage of the project when still defining the RNA to use for the structure determination, we performed many ITC experiments with different RNAs. Below here are two examples of RRM1 and RRM12 with AU-rich RNA very similar to the CUUAUUUG RNA used in the experiments presented in the paper and with which the structure was solved (we know from the structure that it is mainly the UUAUUU which is bound, we added the terminal C and G mainly for NMR assignment purposes). **These data show that the RRM1 binds AU-rich RNA at a K_d larger than 34 μM , RRM12 with a K_d of 1 μM , agreeing very well with the data presented in the paper, showing these K_d s are not dependent on the protein preparation.**

We address the issue on reproducibility by adding these data to the supplement (Fig. S3 D and E) and the K_d s to Table S2. We also added a sentence referring to these data on page 6. Please

note that we discovered a typo on this page – the NMR titration in Fig. 2B was performed with the CUUAAUUUG oligonucleotide.

- 3) In figure S5 panel C, from the fit, but also the shape of the heat rate curve it is clear that the affinity for the single mutant proteins W215F and K197A is weaker compared to the WT. While the affinity for K197A can be successfully fit, the one of W215F is only estimated. For K197A, maybe there a replicate could be done to ensure the reproducibility of the ITC measurement, but **the reproducibility of weaker binding has already been shown in the NMR measurements shown in panel B of figure S5**. There it is very clear that the affinity of these two single mutant proteins is weaker than the wildtype which is the main information we needed to have in order to show that **K197 and W215F are important residues of dnd1 for binding RNA, but that the single mutations are not enough to completely destroy the RRM2 binding pocket**. If the affinity is reduced exactly by 3, 4 or 5-fold in not the important information here.